# DrM: Mastering Visual Reinforcement Learning through Dormant Ratio Minimization

**Guowei Xu**[1][*][†]   **Ruijie Zheng**[2][*]   **Yongyuan Liang**[2][*]   **Xiyao Wang**[2]
**Zhecheng Yuan**[1]   **Tianying Ji**[1]   **Yu Luo**[1]   **Xiaoyu Liu**[2]   **Jiaxin Yuan**[2]   **Pu Hua**[1]
**Shuzhen Li**[1]   **Yanjie Ze**[34]   **Hal Daumé III**[2]   **Furong Huang**[2]   **Huazhe Xu**[145][†]

[1] Tsinghua University    [2] University of Maryland, College Park
[3] Shanghai Jiao Tong University    [4] Shanghai Qi Zhi Institute    [5] Shanghai AI Lab

## Abstract

Visual reinforcement learning (RL) has shown promise in continuous control tasks. Despite its progress, current algorithms are still unsatisfactory in virtually every aspect of the performance such as sample efficiency, asymptotic performance, and their robustness to the choice of random seeds. In this paper, we identify a major shortcoming in existing visual RL methods that is the agents often exhibit sustained inactivity during early training, thereby limiting their ability to explore effectively. Expanding upon this crucial observation, we additionally unveil a significant correlation between the agents' inclination towards motorically inactive exploration and the absence of neuronal activity within their policy networks. To quantify this inactivity, we adopt dormant ratio (Sokar et al., 2023) as a metric to measure inactivity in the RL agent's network. Empirically, we also recognize that the dormant ratio can act as a standalone indicator of an agent's activity level, regardless of the received reward signals. Leveraging the aforementioned insights, we introduce `DrM`, a method that uses three core mechanisms to guide agents' exploration-exploitation trade-offs by actively minimizing the dormant ratio. Experiments demonstrate that `DrM` achieves significant improvements in sample efficiency and asymptotic performance with no broken seeds (76 seeds in total) across three continuous control benchmark environments, including DeepMind Control Suite, MetaWorld, and Adroit. Most importantly, `DrM` is the first model-free algorithm that consistently solves tasks in both the Dog and Manipulator domains from the DeepMind Control Suite as well as three dexterous hand manipulation tasks without demonstrations in Adroit, all based on pixel observations. [1]

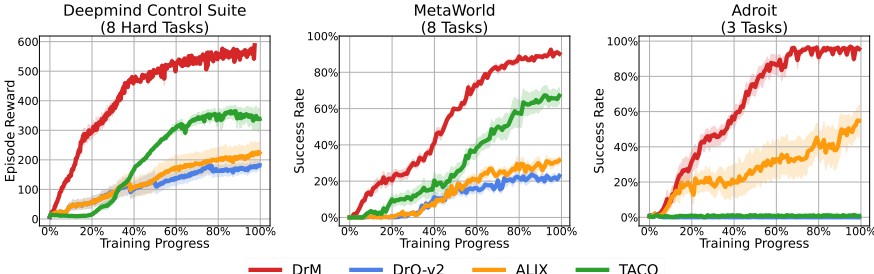

**Figure 1:** Success rate and episode reward as a function of training progress for each of the three domains that we consider (Deepmind Control Suite, MetaWorld, Adroit). All results are averaged over 4 random seeds, and the shaded region stands for standard deviation across different random seeds.

## 1 Introduction

Visual deep reinforcement learning (RL) agents that tackle complex continuous control tasks using high-dimensional pixels are crucial. Recent progress has been made through the incorporation of

---

[*]Equal contribution.

[†]Corresponding author. {`xgw23@mails`, `huazhe_xu@mail`}`.tsinghua.edu.cn`

[1]Please refer to https://drm-rl.github.io/ for experiment videos and benchmark results.

data augmentation (Yarats et al., 2022; 2021; Laskin et al., 2020a), self-supervised representation learning (Zheng et al., 2023; Laskin et al., 2020b; Stooke et al., 2021; Schwarzer et al., 2021; D'Oro et al., 2023), regularization of the temporal difference update (Cetin et al., 2022), and high update-to-data (UTD) ratio (Hiraoka et al., 2022). Nonetheless, the sample efficiency exhibited by these RL agents remains unsatisfactory. To be more specific, visual RL's inability first appears in the face of complex kinematics and a high number of degrees of freedom (DoFs), such as the Dog and Humanoid tasks in the DeepMind Control Suite (Tassa et al., 2018) or dexterous hand manipulation tasks in Adroit (Rajeswaran et al., 2018) without demonstrations. Second, the current leading visual RL agents might get stuck in the local optimum during the learning process under different initial random seeds. The inability to deal with complex systems and the presence of broken random seeds combined pose significant challenges to deploying visual RL agents in real-world applications.

In this paper, we examine the behaviors of visual RL agents at different stages of training. Intriguingly, a recurrent issue we identify by observing the learning agents' behavior is that the agents frequently become motorically inactive during the initial phases of training, hindering the effective exploration of useful behaviors. When the agent is experiencing motor inactivity, we find that the policy neural network also possesses a high rate of inactive neurons, which is defined as dormant neurons (Sokar et al., 2023) in the literature. As the training progresses, the agents' acquisition of new skills is usually accompanied by a decline in the portion of dormant neurons i.e., dormant ratio. Hence, we hypothesize and empirically verify that the dormant ratio acts as an inherent gauge of an agent's activity level, irrespective of the external rewards it receives. Such a connection opens up a new path for balancing between exploration and exploitation in RL agents. Remarkably, this pattern of inactivity in motor skills and neurons mirrors the arousal theory (Harrison & W, 2015; Güzel et al., 2020) in neuroscience, which states that an optimal neural network activity level is essential for enhancing attention, memory, and learning efficiency.

Based on this observation and insight, we propose to train visual RL agents with **D**ormant **r**atio **M**inimization (`DrM`). `DrM` introduces three simple mechanisms to effectively balance between exploration and exploitation while lowering the dormant ratio: a periodical neural network weight perturbation mechanism, a dormant-ratio-based exploration scheduler, and a dormant-ratio-based exploitation mechanism extended from Chen et al. (2021a). Consequently, the agent could emphasize exploration when the dormant ratio is high and shift its focus to exploitation when the dormant ratio is low. `DrM` is easy to implement, computationally efficient, and empirically sample efficient.

`DrM` is evaluated across three different domains, Deepmind Control Suite (Tassa et al., 2018), MetaWorld (Yu et al., 2019), and Adroit (Rajeswaran et al., 2018), including 19 tasks within the realm of locomotion control, tabletop manipulation, and dexterous hand manipulation. Most notably, `DrM` is the **first documented model-free algorithm** that reliably solves complex dog and manipulator tasks, as well as demonstration-free Adroit dexterous hand manipulation tasks from pixels. Furthermore, compared with previous state-of-the-art model-free algorithms, `DrM` is significantly more sample efficient, especially on tasks with sparse rewards. To be precise, our technique requires 70%, 45%, and 60% fewer samples to match the peak asymptotic performance seen in the three baseline methods on the Deepmind Control suite, MetaWorld, and Adroit, respectively. Moreover, in terms of asymptotic performance, our method exhibits improvements of 65%, 35%, and 75% over the best-performing baseline on the Deepmind Control suite, MetaWorld, and Adroit, respectively.

Below, we summarize our key contributions:

1. Through systematic examinations of the dormant ratio within agents performing continuous control tasks, we establish a crucial insight that a decline in the dormant ratio is an early indicator of successful skill acquisition, even before the increase of reward.

2. We introduce a mechanism that periodically perturbs the model weights of the agent, effectively reducing the dormant ratio and hence accelerating skill acquisition.

3. We additionally design a dormant-ratio-based self-adaptive exploration-exploitation scheduler that ensures the agent explores when the dormant ratio is high and exploits its past success when the dormant ratio is low.

4. Extensive experiments on Deepmind Control Suite, MetaWorld, and Adroit show that `DrM` is particularly adept at handling tasks with sparse rewards or complex dynamics, achieving state-of-the-art performance against current leading visual RL baselines. `DrM` is the first

model-free RL algorithm that can reliably solve complex tasks such as Dog, and Manipulator, as well as demonstration-free Adroit dexterous hand manipulation tasks directly from pixels.

## 2 PRELIMINARY

**Visual reinforcement learning.** In visual RL (Kaelbling et al., 1998), the landscape is characterized by the inherent challenge of partial observability when dealing with image inputs, which prompts us to approach the problem as a Partially Observable Markov Decision Process (POMDP) (Bellman, 1957), encapsulated within the tuple $\langle \mathcal{S}, \mathcal{O}, \mathcal{A}, \mathcal{P}, \mathcal{R}, \gamma \rangle$. Here, $\mathcal{S}$ is the state space, $\mathcal{O}$ is the observation space and $\mathcal{A}$ stands for the action space. $\mathcal{P} : \mathcal{S} \times \mathcal{A} \rightarrow \Delta(\mathcal{S})$ defines the state transition kernel, where $\Delta(\mathcal{S})$ is a distribution over the state space. $\mathcal{R} : \mathcal{S} \times \mathcal{A} \rightarrow \mathbb{R}$ denotes the reward function and $\gamma \in [0, 1)$ represents the discount factor. Starting from an initial state $s_0 \in \mathcal{S}$, the overarching objective within this framework is to discover an optimal policy $\pi^* : \mathcal{S} \rightarrow \Delta(\mathcal{A})$ that maximizes the expected cumulative return, formulated as $\mathbb{E}_\pi[\sum_{t=0}^{\infty} \gamma^t r_t]$.

**Dormant Ratio of Neural Network** The notion of dormant neurons, as originally introduced in Sokar et al. (2023), identifies neurons that have become nearly inactive, displaying minimal activation levels. This concept plays an important role in analyzing neural network behavior since networks used in online RL tend to lose their expressive ability.

**Definition 2.1.** *(Sokar et al., 2023) Consider a fully connected layer $l$ with $N^l$ neurons in total. Given an input distribution $\mathcal{D}$, let $h_i^l(x)$ denote the output of neuron $i$ in layer $l$ under input $x \in \mathcal{D}$. The **score** of a neuron $i$ is:*

$$s_i^l = \frac{\mathbb{E}_{x \in \mathcal{D}}|h_i^l(x)|}{\frac{1}{N^l} \sum_{k \in l} \mathbb{E}_{x \in \mathcal{D}}|h_k^l(x)|} \tag{1}$$

*Then we define a neuron $i$ in layer $l$ to be $\tau$-dormant if $s_i^l \leq \tau$.*

**Definition 2.2.** *For a fully connected layer $l$, we denote the number of $\tau$-dormant neurons as $H_\tau^l$. The $\tau$-**dormant ratio** of a neural network $\phi$ can be formally defined as follows:*

$$\beta_\tau = \sum_{l \in \phi} H_\tau^l / \sum_{l \in \phi} N^l \tag{2}$$

## 3 METHOD

In this section, we begin by discussing a key empirical observation: there is a connection between the sharp reduction of an agent's dormant ratio and the agent's skill acquisition in visual continuous control tasks. This is detailed in Section 3.1. Building on top of this crucial insight, in Section 3.2, we introduce our proposed algorithm DrM . In particular, we come up with three simple yet effective mechanisms in DrM such that they aim to not only reduce the agent's dormant ratio but also utilize the calculated dormant ratio to strike a balance between exploration and exploitation.

### 3.1 KEY INSIGHT: DORMANT RATIO AND BEHAVIORAL VARIETY

While previous works Lyle et al. (2022); Sokar et al. (2023) have highlighted that the actor/critic network of RL agents tends to lose expressivity during training, our empirical study offers a unique perspective on visual reinforcement learning for continuous control tasks: the dormant ratio and the agent's behavioral variety are correlated.

To illustrate this, we choose DrQ-v2, a leading model-free RL algorithm that learns directly from pixel observations. In Figure 2, we display the dormant ratio of an agent's policy network, alongside the behaviors learned by the agent during its training on the Hopper Hop task from DeepMind Control Suite as an example. Interestingly, as depicted in this figure, we notice that **a sharp decline in the dormant ratio of an agent's policy network serves as an intrinsic indicator of the agent executing meaningful actions for exploration.** Namely, when the dormant ratio is high, the agent becomes immobilized and struggles to make meaningful movements. However, as this ratio decreases, we observe a clear progression in the agent's mobility, as demonstrated in the figure: starting with crawling, advancing to standing, and ultimately, hopping. We refer the readers to Appendix A and project webpage for more visualizations of the dormant ratio.

Based on these empirical observations, we conclude that the decline in the dormant ratio is closely linked to the agent's initiation of meaningful actions, marking a departure from its prior monotonous

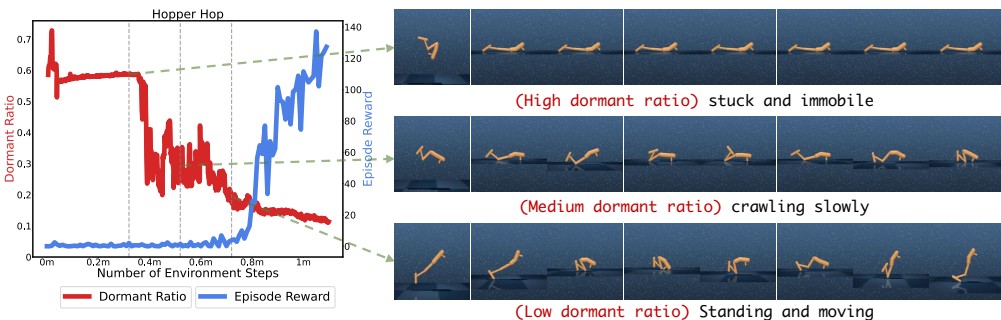

**Figure 2:** (Dormant ratio of a DrQ-v2 agent trained on Hopper Hop task during the first 1M frames): Interestingly, we find that with a declining dormant ratio, the agent incrementally acquires action capabilities. Even though the reward stays minimal during this phase, the dormant ratio provides a more insightful gauge of the agent's initial learning progress than the reward does.

or random behaviors. Interestingly, this shift can happen without a corresponding rise in the agent's rewards. This suggests that the dormant ratio acts as an intrinsic metric, influenced more by the diversity and relevance of the agent's behaviors than by its received rewards, which underscores the value of the dormant ratio as a meaningful metric for understanding the behaviors of visual RL agents.

Motivated by this insight, we aim to utilize dormant ratio as a pivotol tool for balancing exploration and exploitation. Many existing strategies adjust exploration noise based on static factors such as task complexity and training stage. Nonetheless, an agent's performance can fluctuate across tasks and with different initializations, making adjustments based solely on these static factors less efficient and often mandating exntensive, task-specific fine-tuning of hyperparameters. In contrast, customizing exploration noise according to the agent's current performance offers a more flexible and effective approach. While an intuitive approach would be to rely on reward signals, this strategy brings up the following challenges: 1) Reward values definitions vary across different tasks and domains, necessitating domain-specific knowledge for interpretation and hyperparameter tuning. 2) Even within a specific task, rewards might not indicate the agent's underlying learning phase. As depicted in Figure 2, an agent can attain similar rewards regardless of whether it has mastered motion or remains stagnant.

In light of this, the dormant ratio emerges as a more effective metric for adjusting the exploration and exploitation tradeoff, as it faithfully reflects the dynamic changes in the agent's behavior. Our design of `DrM` follows this simple intuition: a higher dormant ratio suggests the need for increased exploration, whereas a lower ratio calls for exploitation. As the dormant ratio captures the intrinsic characteristics of an agent's policy and behaviors, `DrM` is demonstrated to be effective across diverse tasks and domains with minimal hyperparameter tuning required.

## 3.2 DRM : VISUAL REINFORCEMENT LEARNING THROUGH **D**ORMANT **R**ATIO **M**INIMIZATION

As shown in the previous subsection, given a fixed network capacity, it is essential for a visual RL agent to actively reduce its dormant ratio, thereby enabling it to explore the environment through purposeful actions. Driven by this insight, we introduce the three mechanisms of our proposed `DrM` algorithm in detail.

**Dormant-ratio-guided perturbation.** The goal of this mechanism is to perturb the model weights when the RL agent's network displays a high dormant ratio, losing its expressivity. Here, we utilize the perturbation reset method (D'Oro et al., 2023; Ash & Adams, 2020) that employs soft resets, a process that interpolates all the agent's parameters between their prior values and randomly initialized values. This can be expressed with the following equation:

$$\theta_t = \alpha\theta_{t-1} + (1-\alpha)\phi, \phi \sim \text{initializer} \tag{3}$$

Here, $\alpha$ is referred to as the *perturb factor*, $\theta_{t-1}$ indicates the network weights before the reset, $\theta_t$ is the network weight after the reset, and $\phi$ is randomly initialized weights. Note that this is fundamentally different from the approach of NoisyNet (Fortunato et al., 2018b), which is designed to encourage exploration by injecting noise into the model weights at every timestep. Our goal here is to refresh the dormant weights only after a relatively long time interval (every 2e+5 frames). The value of $\alpha$ is controlled by the dormant ratio $\beta$: $\alpha = clip(1 - k\beta, \alpha_{min}, \alpha_{max})$, where $k$ is the perturb rate.

**Awaken exploration scheduler.** We aim to emphasize exploration with a large exploration noise when the dormant ratio is high, and reduce the exploration noise when the dormant ratio is low. Thus, rather than utilizing the linear decay of exploration noise variance in the original DrQ-v2, we introduce a dormant-ratio-based ***awaken exploration scheduler***. Specifically, let $\hat{\beta}$ denote a low dormant ratio threshold. We define the agent as "awakened" when its dormant ratio is below $\hat{\beta}$. Let $t_0$ be the number of timesteps until the agent becomes "awakened" from the start of training. The standard deviation of the exploration noise, $\sigma(t)$, is then defined as:

$$\sigma(t) = \begin{cases} \max\left\{\frac{1}{1+\exp\left(-(\beta-\hat{\beta})/T\right)}, \ \sigma_{\text{linear}}(t-t_0)\right\} & \text{if awakened} \\[2ex] \frac{1}{1+\exp\left(-(\beta-\hat{\beta})/T\right)} & \text{otherwise} \end{cases} \tag{4}$$

Here, $T$ is the exploration temperature hyperparameter. $\sigma_{\text{linear}}(\cdot)$ is the linear schedule of exploration noise defined in DrQ-v2. We visualize the ***awaken exploration scheduler*** in Figure 3 as a function of the dormant ratio. Initially, when the dormant ratio is high, we would like to give the agent a big exploration noise to encourage effective exploration of the environment. As training progresses and the dormant ratio decreases to a relatively low level (below the threshold $\hat{\beta}$), this indicates that the agent should transition from exploration to exploitation.

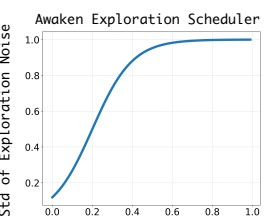

**Figure 3:** Visualization of the awaken exploration scheduler as a function of the dormant ratio $\beta$

**Dormant-ratio-guided exploitation.** Furthermore, we introduce another mechanism that skillfully prioritizes exploitation when the dormant ratio is low. For continuous control tasks using actor-critic algorithms, the critic aims to approximate $r(s,a) + \gamma Q(s', \pi(s'))$. In Ji et al. (2023), it demonstrates that value underestimation often occurs in the early stages of training, when the replay buffer could contain scarce high-quality episodes that the agent has encountered through exploration. In this training stage, $\pi$ is suboptimal, and the $Q$-value is often underestimated due to insufficient exploitation of high-quality samples in the replay buffer. To address this, it proposes to approximate a high expectile of Q values with $V$ function using expectile regression, making the new target value

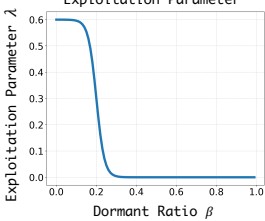

**Figure 4:** Visualization of exploitation hyperparameter as a function of the dormant ratio $\beta$

$$r(s,a) + \gamma[\lambda V(s') + (1-\lambda)Q(s', \pi(s'))], \lambda \in [0,1] \tag{5}$$

As $V$ converges more rapidly than $Q$-values, this mechanism allows the RL agent to quickly exploit its historically successful trajectories without introducing additional overestimation. Here, $\lambda$ serves as the *exploitation hyperparameter*. Higher values of $\lambda$ focus more on exploiting past successes through the fitted V function, the value of the best actions in that state. This emphasis on *exploitation* in our context refers to utilizing the $V$ function to extract more value from historical experiences, aligning with its traditional usage of maximizing rewards based on known information. We introduce a dormant-ratio-guided exploitation technique $\lambda$, which is now defined as a function of the dormant ratio $\beta$:

$$\lambda(\beta) = \frac{\overline{\lambda}}{1 + \exp((\beta - \hat{\beta})/T')} \tag{6}$$

Here, $\overline{\lambda}$ is the maximum exploitation hyperparameter, and $T'$ is the exploitation temperature hyperparameter. $\beta$ and $\hat{\beta}$ represent the dormant ratio and its threshold, as previously defined. In Figure 4, we plot the exploitation hyperparameter $\lambda$ as a function of the dormant ratio $\beta$. When the agent's dormant ratio exceeds the threshold $\hat{\beta}$, a lower $\lambda$ is selected to emphasize exploration. Conversely, when the dormant ratio is low, indicating the agent can perform meaningful actions, a higher $\lambda$ is chosen to prioritize exploitation.

## 4 EXPERIMENT

In this section, we evaluate `DrM` on three visual continuous control benchmarks for both locomotion and robotic manipulation: DeepMind Control Suite (Tassa et al., 2018), MetaWorld (Yu et al.,

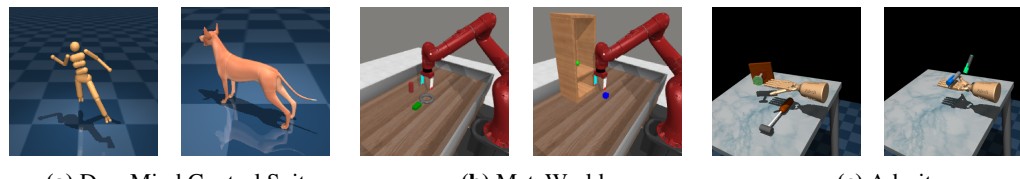

**(a)** DeepMind Control Suite        **(b)** MetaWorld        **(c)** Adroit

**Figure 5:** Three visual continuous control benchmarks to evaluate our proposed algorithm: DeepMind Control Suite, MetaWorld, and Adroit.

2019), and Adroit (Rajeswaran et al., 2018). These environments feature rich visual elements such as textures and shading, necessitate fine-grained control due to complex geometry, and introduce additional challenges such as sparse rewards and high-dimensional action spaces that previous visual RL algorithms such as DrQv2 (Yarats et al., 2022) have been unable to solve.

**Baselines.** We compare our algorithm with the three strongest existing model-free visual RL algorithms: **DrQ-v2** (Yarats et al., 2022), **ALIX** (Cetin et al., 2022), and **TACO** (Zheng et al., 2023). **ALIX** and **TACO** build upon **DrQ-v2**. **ALIX** adds an adaptive regularization to the encoder's gradients to stabilize temporal difference learning from encoders. **TACO** incorporates an auxiliary temporal action-driven contrastive learning objective to learn state and action representations.

**DeepMind control suite.** For Deepmind Control Suite, we evaluate DrM on eight hardest tasks from the Humanoid, Dog, and Manipulator domain, as well as Acrobot Swingup Sparse. The Manipulator domain is particularly challenging due to its sparse reward structure and the long horizon required for skill acquisition, while Humanoid and Dog tasks feature intricate kinematics, skinning weights, collision geometry, as well as muscle and tendon attachment points. This complexity makes these domains extremely difficult for algorithms to learn to control effectively. Following the experimental procedure described by Yarats et al. (2022), we evaluate DrM and all baseline algorithms over 30 million frames of online interaction, while Acrobot Swingup Sparse was run for 6 million frames. Intriguingly, in four dog tasks, we observe that existing baselines encounter a sudden performance decline for some random seeds. We have confirmed this is not due to the checkpoint loading mechanisms, and in contrast, DrM does not exhibit this issue in any of the four tasks. As shown in Figure 6, we note that DrM is the **first documented model-free visual RL algorithm** that is capable of solving both Dog and Manipulator domains in the DeepMind Control Suite using pixel observations. Additionally, we notice that the variation across different random seeds, as indicated by the shaded areas in our results, is considerably smaller for DrM compared to baseline algorithms. This reduced variation implies that DrM is more robust to different random initializations. In contrast, baseline algorithms frequently experience issues with broken seeds, where the agent fails to acquire any meaningful behaviors and receives consistently low rewards throughout the training process.

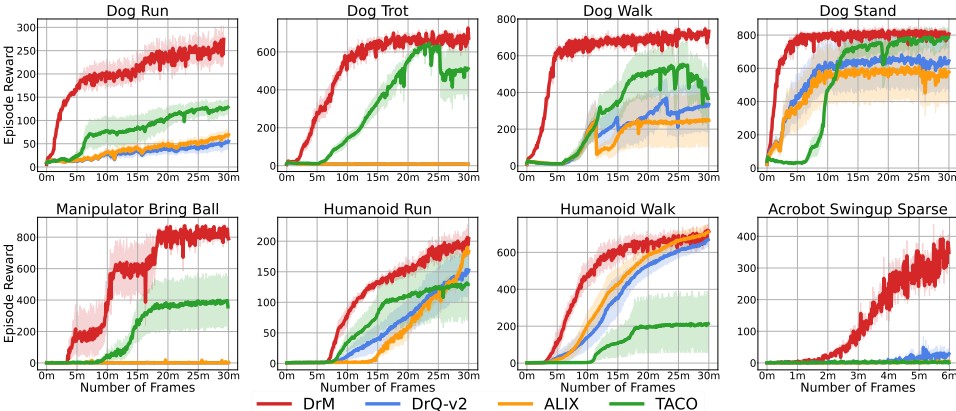

**Figure 6:** Performance of DrM against baseline algorithms **DrQ-v2**, **ALIX**, and **TACO** on Deepmind Control suite. All results are averaged over 4 random seeds, and the shaded region stands for standard deviation across different random seeds.

**MetaWorld.** As shown in Figure 7, we evaluate DrM and baselines on eight challenging tasks including 4 very hard tasks with dense rewards following prior works and 4 medium tasks with sparse

success signals. Consistently across the spectrum of tasks within MetaWorld, our method outperforms other visual RL baselines, which demonstrates the significantly improved sample efficiency of `DrM`. Especially in more challenging scenarios featuring only sparse task completion rewards, existing visual RL baselines struggle to find a good policy, while `DrM` shines by achieving success rates on par with those using dense reward signals. This underscores the remarkable advantages brought by dormant-ratio-based exploration when dealing with tasks with sparse rewards.

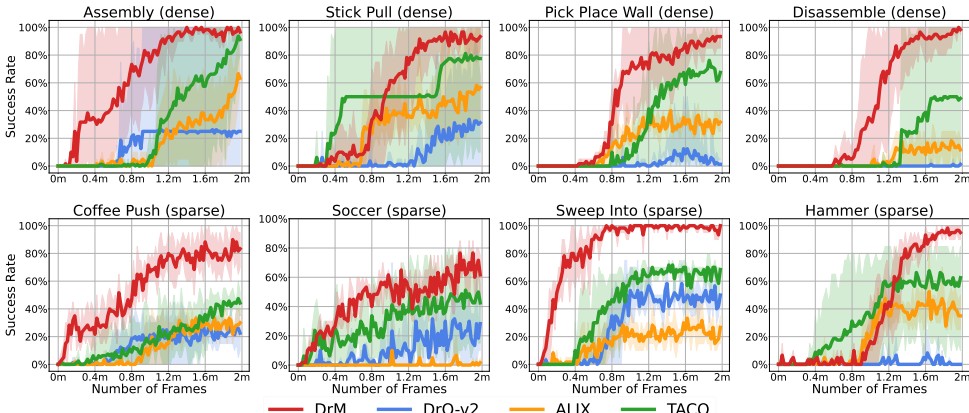

**Figure 7:** Success rate for `DrM` and baselines on MetaWorld including 4 very hard tasks with dense rewards and 4 medium tasks with spare rewards. All results are aggregated over 4 random seeds, with shaded areas representing the standard deviation across seeds. Notably, our method demonstrates significantly higher sample efficiency, especially in tasks with sparse rewards.

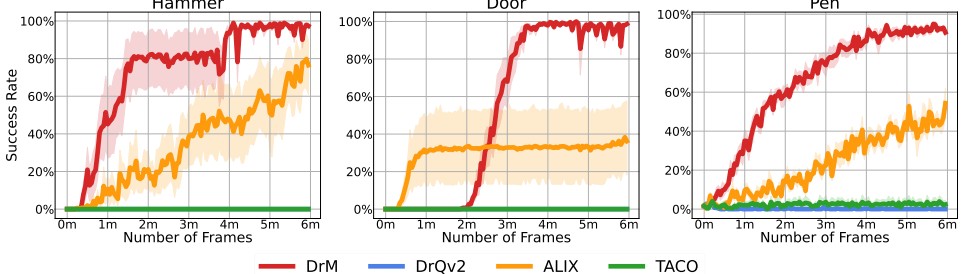

**Figure 8:** Performance of `DrM` against baseline algorithms **DrQ-v2**, **ALIX**, and **TACO** on Adroit. All results are averaged over 4 random seeds, and the shaded region stands for standard deviation across different random seeds.

**Adroit.** In Figure 8, we also evaluate `DrM` on the Adroit domain, focusing on three dexterous hand manipulation tasks: Hammer, Door, and Pen, which requires controlling a robotic hand with 24 degrees of freedom. For additional task details, we refer readers to Rajeswaran et al. (2018). Given the task's high-dimensional action space and intricate physics, previous reinforcement learning algorithms have faced significant challenges, especially when learning from pixel observations. Notably, `DrM` is the **first documented model-free visual RL algorithm** that is capable of reliably solving tasks in the Adroit domain **without expert demonstrations**.

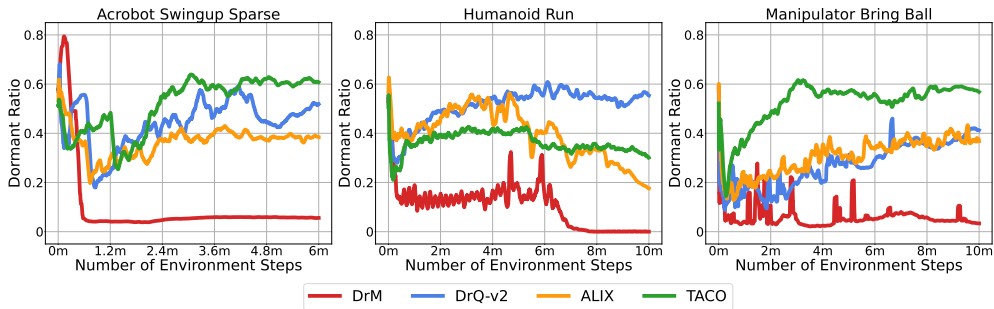

**Figure 9:** Dormant ratio of our method `DrM` and other baseline algorithms on three DMC tasks.

**Dormant Ratio Analysis** In this section, we conduct a detailed analysis and comparison of the dormant ratio changes during the training process of `DrM` and three baseline algorithms. We carry out experiments in three visual DMC tasks, and the experimental results are shown in Figure 9. From this figure, we observe that as training progresses, the dormant ratio of `DrM` rapidly decreases, indicating that our method effectively minimizes the dormant ratio. In comparison, other exisiting baselines all fail to effectively reduce the dormant ratio. This also explains why our approach exhibits high sample efficiency and performance.

**Ablation Study** We conduct ablation studies on the Adroit environment to evaluate the contribution of each component to our method, i.e., dormant-ratio-guided perturbation, awaken exploration, and dormant-ratio-guided exploitation. Additionally, to show that the dormant ratio plays a crucial role in integrating these three components, we also compare with a baseline where we use fixed parameters for the three mechanisms without being guided by the dormant ratio. (i.e., Drg perturbation with perturb factor $\alpha$ fixed, fixed linear exploration schedule, and Drg exploitation with exploitation parameter $\lambda$ fixed.)

The experiment results are shown in Figure 10. From the results, we find that all three components are necessary to achieve the best results.

We observe that after removing the dormant-ratio-guided exploitation (DrM w/o Drg Exploitation), the final success rate decreased by nearly 20%, while eliminating either the dormant-ratio-guided perturbation (DrM w/o Drg Perturbation) or the awaken exploration (DrM w/o Awaken Exploration) lead to a decline of close to 40%, highlighting the importance of each component. In our ablated version without dormant-ratio-guided perturbation, the agent only converges to a suboptimal policy, reaching a success rate of just about 40%. This is likely due to the fact that without the awaken exploration, the agent lacks sufficient exploration, making it easy to get stuck in a sub-optimal policy. Additionally, when removing the dormant-ratio-guided exploitation component, the agent lacks the ability to exploit its past success, and there fore exhibits a significantly slower learning curve.

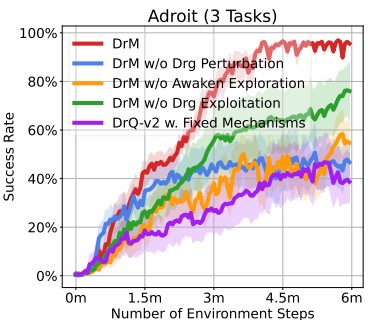

**Figure 10:** (**Ablation**) Aggregated results over Adroit domain. Drg represents Dormant-ratio-guided.

## 5 RELATED WORK

**Visual reinforcement learning.** Visual reinforcement learning (RL) faces substantial challenges when training agents to make decisions based on pixel observations. Within this domain, two primary categories of approaches have emerged: model-based and model-free methods. Model-based methods (Hansen et al., 2022; Hafner et al., 2020; 2021; 2019; Lee et al., 2020; Hafner et al., 2023) accelerate visual RL by learning world models of the environment. On the other hand, model-free methods have made significant strides in improving data efficiency. These advancements include auxiliary losses, such as the contrastive objective in CURL (Laskin et al., 2020b), ATC (Stooke et al., 2021) for state representations, TACO (Zheng et al., 2023) for learning state and action representations through mutual information, and self-prediction representations in SPR (Schwarzer et al., 2021) and SR-SPR (D'Oro et al., 2023). Data augmentation techniques, exemplified by RAD (Laskin et al., 2020a), DrQ (Yarats et al., 2021), and its enhanced version DrQv2 (Yarats et al., 2022), have been instrumental in enabling robust learning directly from pixel data, effectively bridging the gap between state-based and image-based RL. Additionally, regularization methods such as A-LIX (Cetin et al., 2022) have been introduced to mitigate catastrophic self-overfitting by providing adaptive regularization to convolutional features. Furthermore, strategies such as scaling network sizes (Schwarzer et al., 2023), high update-to-data (UTD) ratios (D'Oro et al., 2023) and ensemble Q (Chen et al., 2021b; Hiraoka et al., 2022) have been explored to enhance sample efficiency in visual RL. TD-MPC (Hansen et al., 2022) merges the advantages of model-based and model-free methods through temporal difference learning. V-MPO (Song et al., 2020b), an on-policy adaptation of MPO (Song et al., 2020a), exhibits high asymptotic performance on challenging pixel-control tasks (Tassa et al., 2018). These various techniques collectively represent the state-of-the-art in visual RL, addressing the multifaceted challenges associated with decision-making from raw visual input. However, our proposed framework differs in that we address the sample efficiency challenge from

the perspective of dormant ratio. We propose more effective `DrM` that achieves superior performance than prior model-free baselines.

**Loss of expressivity of deep RL.** In deep RL, there is a growing body of evidence suggesting that neural networks tend to lose their capacity and expressiveness for fitting new targets over time and ultimately harm their final performance. To alleviate this issue, Lyle et al. (2022) and Kumar et al. (2021) primarily focus on adjusting the learned feature values. Nikishin et al. (2022) shed light on the primacy bias when training on early data, which can impede further learning progress. Their proposal involves periodic parameter reinitialization for the last few layers while keeping the replay buffer unchanged. Lyle et al. (2023) aims to identify that the loss of plasticity is fundamentally influenced by the curvature of the loss landscape. Additionally, the dormant neuron phenomenon, as demonstrated by Sokar et al. (2023) prompts the development of ReDo, a method aimed at reducing dormant neurons and preserving network expressivity during training. Nikishin et al. (2023) introduces plasticity injection, a minimalistic intervention that temporarily freezes the current network and leverages newly initialized weights to facilitate continuous learning. These diverse approaches collectively address the issue of expressivity loss in deep RL, offering insights and methods to enhance computational efficiency and continual learning capabilities in deep RL algorithms. In our paper, we leverage the dormant ratio to gain valuable insights and interpretability into agent behavior in visual RL. We introduce a novel perturbation technique and exploration strategy based on the dormant ratio for addressing visual continuous control tasks.

**Exploration in RL.** Efficient exploration remains a substantial challenge in online RL, particularly in high-dimensional environments with sparse rewards. Based on different key ideas and principles, exploration strategies can be classified into two major categories. The first category is uncertainty-oriented exploration (Jin et al., 2020; Ménard et al., 2021a;b; Kaufmann et al., 2021; Wang et al., 2023), which often employs techniques such as the upper confidence bound (UCB) (Auer, 2002) to capture value estimate uncertainty to guide exploration. Another category is intrinsic motivation-oriented exploration, which encourages agents to explore by maximizing intrinsic rewards. These rewards are often based on prediction errors (Houthooft et al., 2016; Pathak et al., 2017; Burda et al., 2019; Sekar et al., 2020; Badia et al., 2020) or count-based state novelty (Bellemare et al., 2016; Tang et al., 2017; Ostrovski et al., 2017), motivating the agent to visit states with high prediction errors or the unexplored states. A close idea is exploration by maximizing state entropy as an intrinsic reward (Lee et al., 2019; Hazan et al., 2019; Mutti et al., 2022; Yang & Spaan, 2023). Exploration methods have proven effective in enhancing sample efficiency in vision-based RL. RE3(Seo et al., 2021) utilizes a fixed random encoder to obtain a stable state entropy estimate, along with a value-conditioned extension proposed in Kim et al. (2023). MADE (Zhang et al., 2021) introduces an adaptive regularization that maximizes deviation from explored regions, while BEE (Chen et al., 2021a) leverages past successes to capitalize on fortuitous circumstances. Closely relevant techniques involve injecting noise into action (Wawrzynski, 2015; Lillicrap et al., 2016) or parameter spaces (Rückstieß et al., 2010; Sehnke et al., 2010; Fortunato et al., 2018a; Plappert et al., 2018). Furthermore, strategies that dynamically adjust exploration noise based on factors like agent performance, environmental complexity, and training stage have shown promise in Amos et al. (2021); Yarats et al. (2022). Our method distinguishes itself by directly perturbing the model weights of the agent to reduce the dormant ratio and design a dormant-ratio-guide exploration technique to improve exploration efficiency.

# 6 CONCLUSION

In this paper, we introduce a highly efficient online RL algorithm, `DrM` , which resolves the most complex visual control tasks that previous models failed to tackle, setting a new benchmark in both sample and time efficiency. Looking ahead, we perceive two main avenues for future RL exploration research. Firstly, the dormant ratio's interpretability is a captivating aspect, and subsequent research could delve into why it has a significant correlation with the diversity and significance of an agent's action, from a theoretical standpoint. Secondly, as the dormant ratio delivers a more precise depiction of an agent's early learning outcomes compared to rewards, it could be used in unsupervised RL. Additionally, although this work primarily concentrates on continuous control, the three key mechanisms we propose for `DrM` could well be adapted for discrete action tasks on DQN/Efficient Rainbow algorithms with some minor adjustments. We are confident that the dormant ratio's value

extends beyond our current understanding, and that its strategic application could greatly enhance the performance of future visual reinforcement algorithms.

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

# A MORE VISUALIZATION RESULTS OF DORMANT RATIO

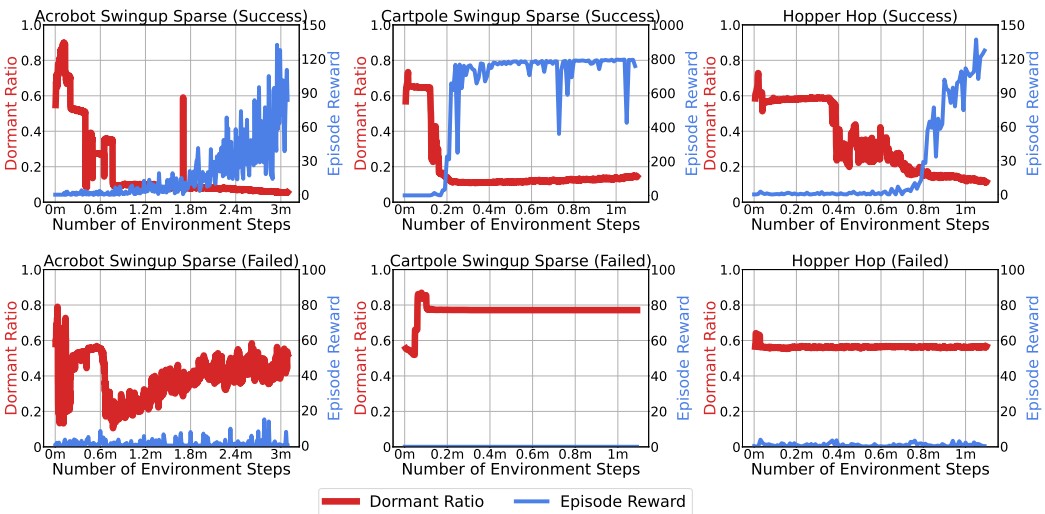

**Figure 11:** Analysis of the dormant ratio in successful vs. broken seeds reveals distinct behavior patterns. In a successful seed, a decreasing dormant ratio allows the agent to effectively explore the environment and learn skills. Conversely, in a broken seed, the agent becomes immobile and fails to discover meaningful motions.

# B TIME EFFICIENCY OF DRM

To assess the algorithms' speed, we measure their frames per second (**FPS**) on the same DeepMind Control Suite task, Dog Walk, using an identical Nvidia RTX A5000 GPU. As Figure 12 shows, while achieving significant sample efficiency and asymptotic performance, DrM only slightly compromises wall-clock time compared to **DrQ-v2**. Compared with two other baselines, DrM is roughly as time-efficient as **ALIX** and about three times faster than **TACO**, which needs a batch size four times larger than that of **DrQ-v2** to compute its temporal contrastive loss.

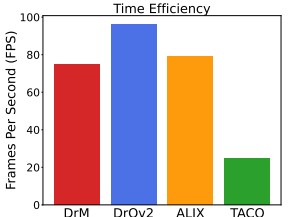

**Figure 12:** Comparison of time-efficiency

## C  IMPLEMENTATION DETAILS

In this section, we describe the implementation details of `DrM`. We have built `DrM` upon the publicly available source code of DrQ-v2. Subsequently, we present the pseudo-code outlining our approach.

### C.1  DORMANT RATIO CALCULATION

In this subsection, we demonstrate how the dormant ratio is calculated in Algorithm 1.

---
**Algorithm 1** Dormant Ratio Calculation
---
1: **procedure** CAL_DORMANT_RATIO(model, inputs, $\tau$-dormant threshold)
2:     Initialize counters: total_neurons, dormant_neurons
3:     Operate a forward propagation: model(inputs)
4:     **for** each module in model **do**
5:         **if** module is Fully Connected Layer **then**
6:             output  =  average over the batch (|output of the forward propagation|)
7:             average_output  =  average over the neurons (output)
8:             dormant_neurons + =  count ( output < average_output $\times \tau$-dormant threshold )
9:             total_neurons + =  neurons in this layer
10:        **end if**
11:    **end for**
12:    **return** dormant ratio = dormant_neurons / total_neurons
13: **end procedure**

---

### C.2  DORMANT-RATIO-GUIDED PERTURBATION

In this subsection, we demonstrate how the perturbation is performed based on the dormant ratio in Algorithm 2.

---
**Algorithm 2** Dormant-ratio-guided Perturbation
---
1: **procedure** PERTURB(network, optimizer, perturb_factor)
2:     Create a new network new_net which has the same shape as the original network
3:     Initialize weights of new_net
4:     **for** each layer and parameter in the network **do**
5:         **if** layer is Fully Connected Layer **then**
6:             Compute noise as: new_net $\times$ (1 − perturb_factor)
7:             Update parameter with: net $\times$ perturb_factor + noise
8:         **end if**
9:     **end for**
10:    Reset the state of the optimizer
11:    **return** updated network, optimizer
12: **end procedure**

---

### C.3 AWAKEN EXPLORATION SCHEDULER

In this subsection, we demonstrate how the awaken exploration scheduler is performed in Algorithm 3.

---

**Algorithm 3** Awaken Exploration Scheduler

---

1: Initialize awaken_step to None
2: **function** STDDEV(step)
3:     **if** awaken_step is None **then**
4:         **return** dormant_stddev
5:     **else**
6:         linear_stddev = linear_schedule(step − awaken_step)
7:         **return** max(dormant_stddev, linear_stddev)
8:     **end if**
9: **end function**
10: **function** UPDATE_AWAKEN_STEP(step)
11:     **if** awaken_step is None **and** dormant_ratio < dormant_ratio_threshold **then**
12:         awaken_step ← step
13:     **end if**
14: **end function**

---

### C.4 DORMANT-RATIO-GUIDED EXPLOITATION

In this subsection, we demonstrate how the dormant-ratio-guided exploitation is performed in Algorithm 4.

---

**Algorithm 4** Dormant-ratio-guided exploitation

---

1: **function** UPDATE_VALUE_NETWORK(obs, action)
2:     $Q1, Q2$ = critic(obs, action)
3:     $Q = \min(Q1, Q2)$
4:     $V = V_{net}(obs)$
5:     error = $V - Q$
6:     sign = $\begin{cases} 1 & \text{if error} > 0 \\ 0 & \text{otherwise} \end{cases}$
7:     weight = $(1 - \text{sign})$expectile + sign$(1 - \text{expectile})$
8:     value_loss = mean(weight × error$^2$)
9:     Update value network using value_loss
10: **end function**
11: **function** CAL_TARGET_Q(next_obs, reward, discount)
12:     action distribution = actor(next_obs, awaken exploration scheduler)
13:     Sample next_action from the distribution with clipping
14:     target_Q1, target_Q2 = critic_target(next_obs, next_action)
15:     target_V_explore = min(target_Q1, target_Q2)
16:     target_V_exploit = $V_{net}$(next_obs)
17:     target_V = $\lambda$ × target_V_exploit + $(1 - \lambda)$ × target_V_explore
18:     target_Q = reward + (discount × target_V)
19:     **return** target_Q
20: **end function**

---

# D    HYPERPARAMETERS

## D.1    HYPERPARAMETERS IN DRM

We summarize all the hyperparameters of DrM in Table 1. While we are trying to keep the settings identical for each of the task, there are a few specific deviations of DrM hyperparameters for some tasks. Additionally, in D.2, we demonstrate the performance of DrM with a single set of hyperparameters across these tasks.

**Hammer, Pen, Door of Adroit**: Exploitation expectile 0.7
**Dog [Stand, Walk, Run], Humanoid Run, Coffee Push & Soccer:** Maximum perturb factor $\alpha_{max} = 0.6$

| Parameter | Setting |
|---|---|
| Replay buffer capacity | $10^6$ |
| Action repeat | 2 |
| Seed frames | 4000 |
| Exploration steps | 2000 |
| $n$-step returns | 3 |
| Mini-batch size | 256 |
| Discount $\gamma$ | 0.99 |
| Optimizer | Adam |
| Learning rate | $8 \times 10^{-5}$ (DeepMind Control Suite) |
| | $10^{-4}$ (MetaWorld & Adroit) |
| Agent update frequency | 2 |
| Soft update rate | 0.01 |
| Features dimension | 100 (Humanoid & Dog) |
| | 50 (Others) |
| Hidden dimension | 1024 |
| $\tau$-Dormant ratio | 0.025 |
| Dormant ratio threshold $\hat{\beta}$ | 0.2 |
| Minimum perturb factor $\alpha_{min}$ | 0.2 |
| Maximum perturb factor $\alpha_{max}$ | 0.9 |
| Perturb rate $k$ | 2 |
| Perturb frames | 200000 |
| Linear exploration stddev. clip | 0.3 |
| Linear exploration stddev. schedule | linear(1.0, 0.1, 2000000) (DeepMind Control Suite) |
| | linear(1.0,0.1,300000) (MetaWorld & Adroit) |
| Awaken exploration temperature $T$ | 0.1 |
| Target exploitation parameter $\hat{\lambda}$ | 0.6 |
| Exploitation temperature $T'$ | 0.02 |
| Exploitation expectile | 0.9 |

**Table 1:** A default set of hyper-parameters used in our experiments.

Note: For learning rate, feature dimension, and linear exploration schedule, we simply follow the standard of DrQ-v2, which has a separate setting for hard DMC tasks (lower learning rate, larger feature dimensionality, longer exploration schedule). These three hyperparameters are not introduced by DrM , and we do not do any tuning on these three.

## D.2 Performance with one set of DrM hyperparameters

Here we show the performance of DrM on tasks where we use a single set of hyperparameters instead of the domain-specific ones.

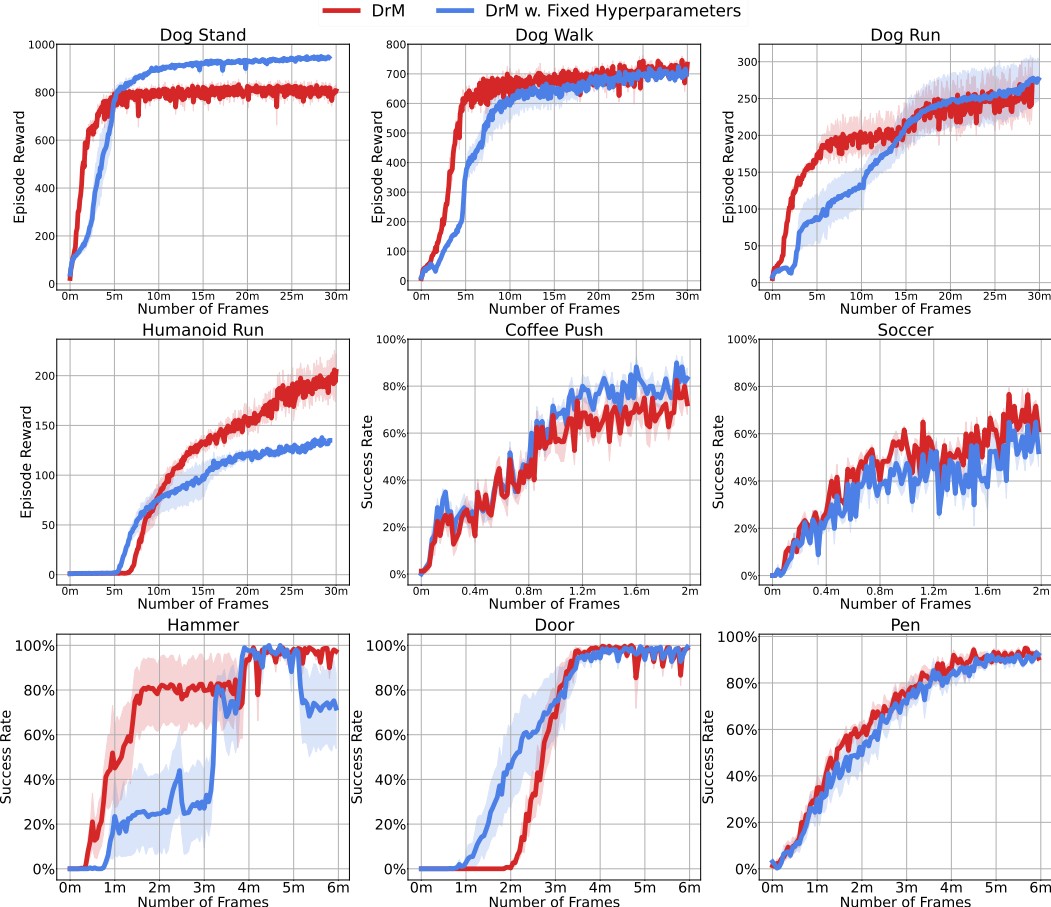

**Figure 13:** Comparison of DrM against DrM with fixed hyperparameters nine tasks where we used the non-default hyperparameter settings.

In general, we find that applying DrM with the default hyperparameter setting, using a unified set of hyperparameter off-the-shelf has a decently great performance across all tasks. But for some tasks such as Humanoid Run in DMC and Hammer in Adroit, some additional hyperparameter tuning would be beneficial to get further performance gain.

# E   DrM versus Intrinsic Reward Based Exploration Techniques

While the primary advantage of our method (`DrM`) lies in employing the dormant ratio to guide the agent's exploration-exploitation tradeoff, encouraging more active exploration of the environment, `DrM` demonstrates remarkable performance on the most challenging tasks, particularly those involving complex environmental dynamics and sparse rewards. Here, we compare `DrM` with an intrinsic reward-based exploration approach, which also aims for encouraging the exploration of a RL agent. For this comparison, we select Random Network Distillation (RND), a popular and widely used technique in intrinsic reward based exploration.

For RND, we implement RND on top of DrQ-v2 and compare its performance against `DrM` on three Adroit tasks. We introduce a random encoder $f$ whose architecture is same as the encoder in DrQ-v2, and we introduce a predictor network $g$ with the same architecture. The predictor network is then trained to predict representations from a random encoder $f$ given the same observations, i.e., minimizing $\epsilon = \|f(s_i) - g(s_i)\|^2$. We use prediction error $\epsilon$ as an intrinsic reward and learn a policy that maximizes $r^{\text{total}} = r^e + \rho r^i$. We perform hyperparameter search over the weight $\rho \in \{0.1, 1.0, 10.0\}$ on the Pen task and and then use the best hyperparameter ($\rho = 1.0$) for the three Adroit tasks.

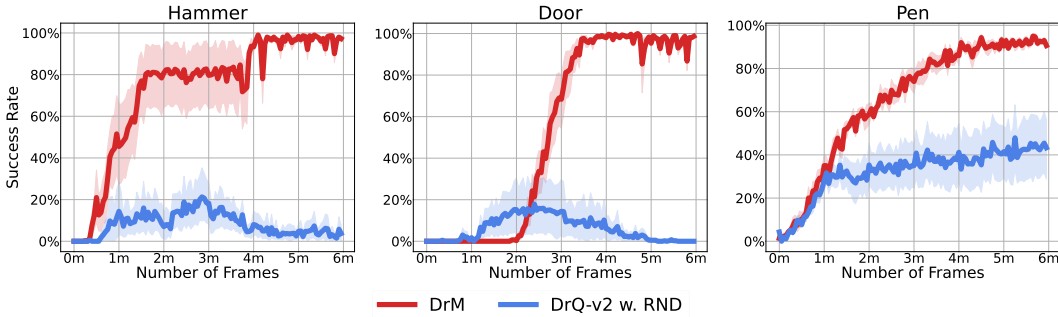

**Figure 14:** Comparison of `DrM` against an intrinsic exploration baseline (DrQ-v2 with RND) on three Adroit tasks.

As shown in Figure 14, an additional intrinsic exploration mechanism such as RND could indeed improve DrQ-v2's performance. (DrQ-v2 achieves 0% success rate across all tasks.) However, adding only an intrinsic exploration mechanism is still insufficient for the agents to discover the optimal policies. The performance gap again demonstrates the significance of `DrM`, which uses the dormant ratio to guide the agent's exploration-exploitation tradeoff. Furthermore, in principle, an RND-like intrinsic exploration technique could also be combined with `DrM` to further boost exploration. This integration could be implemented as a separate mechanism or as a replacement for the awaken exploration schedule in `DrM`. We could also use a similar strategy as in `DrM`, using the dormant ratio to control the magnitude of the intrinsic noise $\rho$. While we encourage future research in this direction, such exploration falls outside the scope of the current work.

## F    COMPARISON WITH REDO

To compare our approach with ReDo (Sokar et al., 2023), which only resets the weights of dormant neurons, we conducted experiments in three different environments on MetaWorld . In Figure 15, it can be observed that our method significantly outperforms the approach of resetting only dormant neurons and `DrM` with ReDo. We speculate that this improvement is due to the positive impact of resetting non-dormant neurons on exploration. Additionally, our use of the dormant ratio to guide exploration strategy distinguishes our approach from previous works.

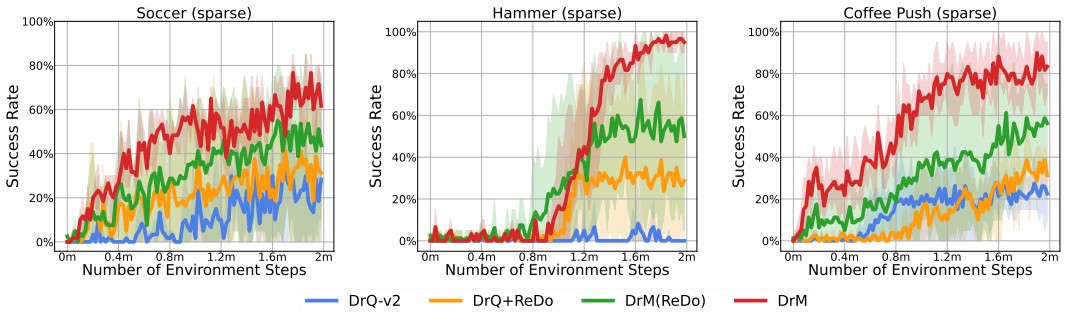

**Figure 15:** Comparison of `DrM` against ReDo (Sokar et al., 2023)

## G    DETAILED ABLATION RESULTS

In this section, we present the results of additional ablation studies for `DrM` in addition to Figure 10. In particular, we conduct additional ablation studies on two MetWorld tasks, and we show the ablation study in three Adroit tasks separately instead of an aggregated plot.

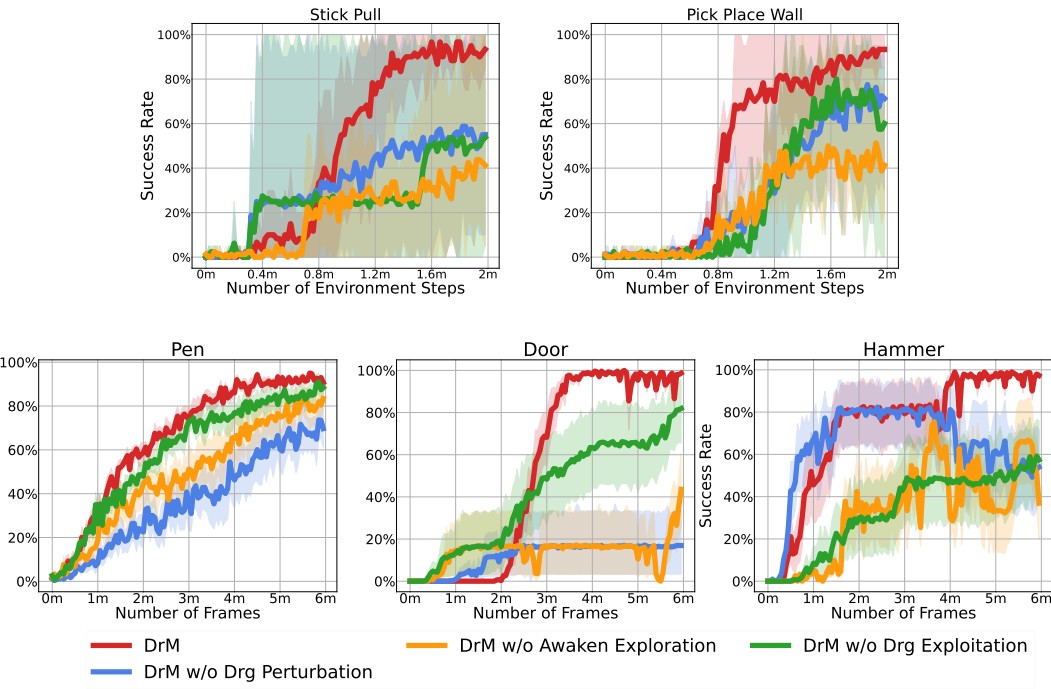

**Figure 16:** Additional ablation studies: Drg represents Dormant-ratio-guided.

# H  AN ADDITIONAL ABLATIONON STUDY ON DORMANT-RATIO-GUIDED EXPLORATION

To justify our design choice of dormant-ratio-guided exploration, here we have conducted an additional ablation study on the Adroit domain, where we compare `DrM` with a baseline such that it sets a maximal exploration noise (i.e., std $= 1$) before awakening and then using the linear schedule afterward.

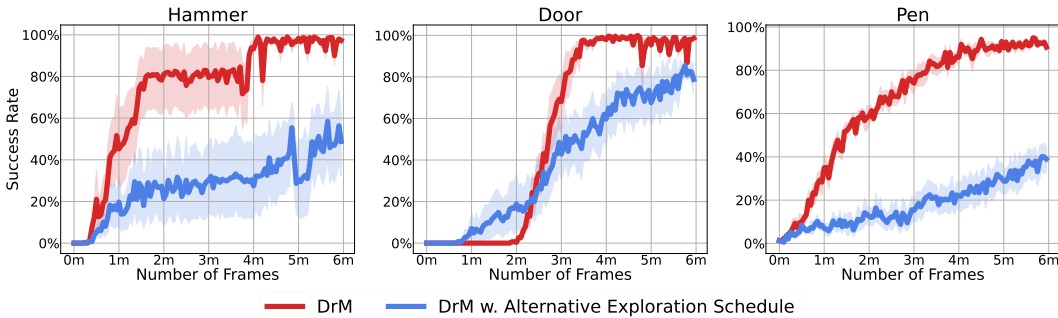

**Figure 17:** Comparison of `DrM` with an alternative exploration strategy

As shown from Figure 17, we find that using the maximal exploration noise instead of our adaptive adjustment of exploration noise based on the dormant ratio results in significant performance degradation across all three Adroit tasks. This further justifies our design choice of the Dormant-ratio-guided exploration mechanism.

# I  RESULTS WITH MORE RUNS

Considering the variability in results introduced by random seeds and for statistical rigor, we conducted 10 runs of experiments for both DrQ-v2 and `DrM` across multiple MetaWorld environments, following the suggestion by Patterson et al. (2023). This was done to compare the performance of the algorithms across a broader range of seeds. It is evident that our algorithm is not sensitive to the randomness of seeds, and it consistently maintains a significant performance lead over baseline algorithms.

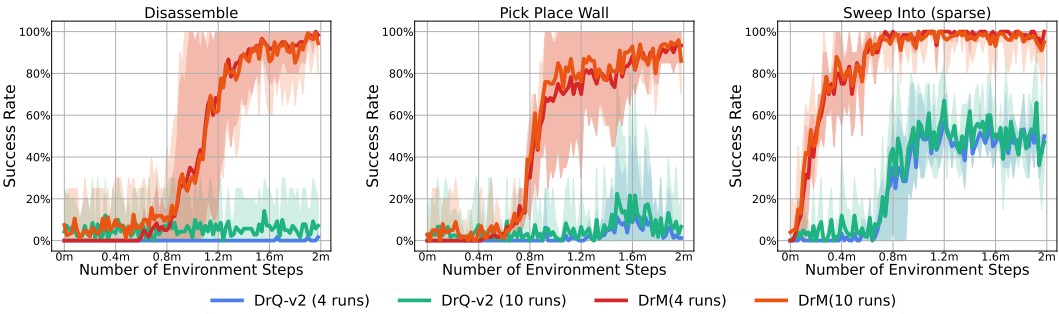

**Figure 18:** 10 runs of `DrM` vs. DrQ-v2 on three Metaworld Tasks.

# J  EFFECT OF SHRINK-AND-PERTURB

Regarding the potential complementary effects of "dithering" exploration introduced by the awaken scheduler and deeper exploration induced by network resets, we perform experiments by replacing reinitialized perturbations with the original initialization parameters. The results, depicted in the following figures, demonstrate that 1-shrink perturbations caused only a 10% decrease in performance in the Sweep-Into while maintaining comparable performance in Stick-Pull. This suggests that the combination of these exploration strategies may be complementary, providing a nuanced understanding of their interplay.

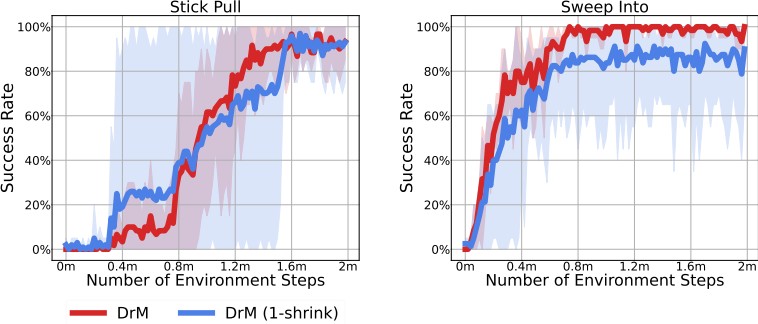

## K    FURTHER DISCUSSION OF DORMANT RATIO

In this section, we aim to evaluate an alternative hypothesis regarding the dormant ratio and `DrM`. The hypothesis is that dormant ratio measures the change from frame to frame and by minimizing it, `DrM` promotes the agent to maximize the velocity of change between frames, thereby encouraging exploration and yielding good empirical results. Through three experiments, we demonstrate that this is not the case. Instead, as argued in the main paper, the dormant ratio should be an intrinsic indicator of the agent's activity level, rather than merely reflecting the speed of change in observations from external environments.

First, we performed an additional experiment using the DMC-Generalization Benchmark (Hansen & Wang, 2021; Yuan et al., 2023). Unlike standard DMC tasks where the agents have a static background, DMC-Generalization introduces a dynamic element by inserting a video clip into the background. If a low dormant ratio truly corresponded to significant frame-to-frame changes, then we would expect the dormant ratio in DMC-Gen to be low throughout the training process. However, our findings contradict this assumption. As demonstrated in Figure 19, the pattern of the dormant ratio in DMC-Gen mirrors that of ordinary DMC tasks. Initially, the agent's dormant ratio remains high, instead of being consistently low throughout the training, challenging the hypothesis that dormant ratio is merely a reflection of rapid frame-to-frame changes.

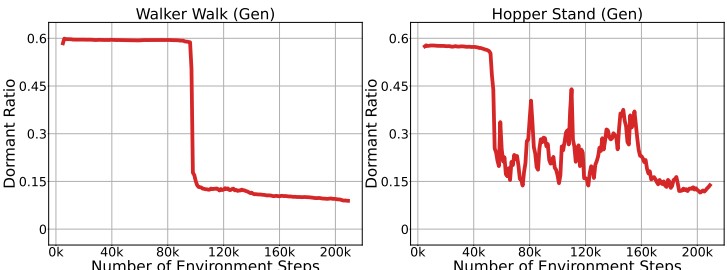

**Figure 19:** The dormant ratio of `DrM` in DMC-Generalization Benchmark.

Next, to further investigate whether using the difference between frames as an intrinsic reward for optimization is effective, we conducted an additional experiment on three Adroit tasks. The hypothesis suggests that by using the difference between consecutive observation frames as an intrinsic reward, an agent could achieve performance comparable to `DrM` by maximizing this reward. However, our empirical tests on three Adroit tasks indicate otherwise. For these experiments, we defined the intrinsic reward $r_t^{\mathrm{i}}$ as the difference between the first and third frame of the agent's observations at timestep $t$. (Here, same as `DrM`, we follow the standard practice to use a stack of three consecutive image frames as the agent's observation at each timestep.) For our experiments, we calculated the L1 difference between the first and third frames. We then normalized this intrinsic reward using the running mean and standard deviation and trained a policy to maximize the total reward $r^{\mathrm{total}} = r^{\mathrm{e}} + r^{\mathrm{i}}$.

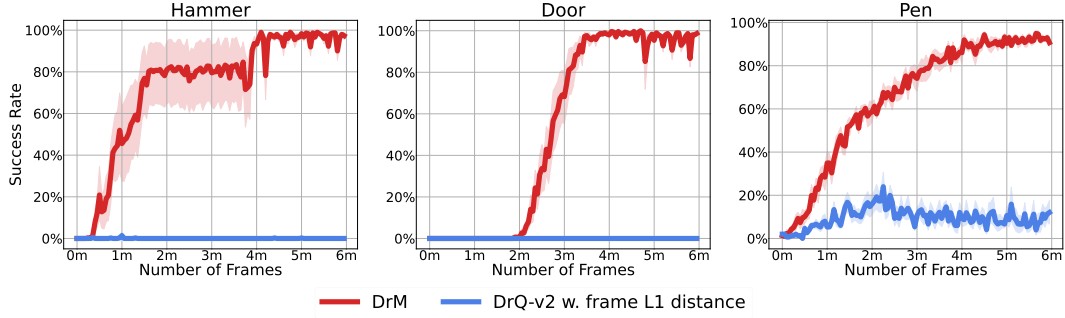

**Figure 20:** `DrM` vs. DrQ-v2 with intrinsic reward defined as L1 difference between consecutive frames

In Figure 20, we show the performance of `DrM` against such baseline. As shown from the plot, simply maximizing the difference between frames clearly cannot solve the three tasks.

Finally, it's important to note that if `DrM` solely focused on maximizing the difference between frames, it would likely struggle with tasks that require minimal motion, such as Acrobot Swingup, Humanoid Stand, and Dog Stand. In these tasks, excessive motion could lead to a low reward. Here we conducted an additional experiment on Cartpole Balance Sparse, a task that also necessitates reduced motion for maintaining balance. As illustrated in Figure 21, despite its simplicity, `DrM` continues to perform well when compared to baseline algorithms. This further indicates that `DrM`'s effectiveness is not merely a result of maximizing frame-to-frame differences.

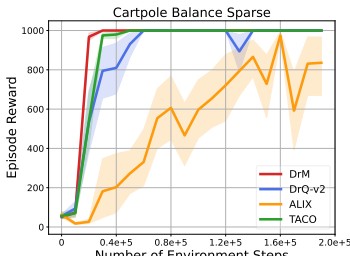

**Figure 21:** `DrM` against baseline algorithms on Cartpole Balance Sparse

