# OpenReview forum: "DrM: Mastering Visual Reinforcement Learning through Dormant Ratio Minimization"
_ICLR.cc/2024/Conference — ICLR 2024 spotlight_

### Official Review · Reviewer_Jxw3 · 2023-10-30

**Soundness:** 3 good
**Presentation:** 3 good
**Contribution:** 3 good
**Rating:** 6
**Confidence:** 4

**Summary:**

This paper is inspired by recent work on the "dormant ratio", a measure of how many neurons are inactive in a deep RL agent's neural net. Previous work (Sokar et al., 2023) showed that a high dormant ratio is detrimental to learning, but this paper flips things around by considering the converse, interpreting a declining dormant ratio as a proxy for learning progress. The authors propose three adaptive hyperparameter scheduling mechanisms based on this idea. They evaluate the resultant algorithm, named DrM, in three visual continuous control domains, where it performs more stably and achieves better asymptotic performance than three other recent methods.

**Strengths:**

The greatest strength of the paper is the significance of the results. DrM is the clear winner on *all* of the tasks reported on in the paper, and in some domains (particularly Adroit) it achieves a step change improvement. I like that the authors have considered three separate domains (DeepMind Control Suite, MetaWorld and Adroit), with multiple tasks from each.

The main idea of treating the dormant ratio as a progress proxy, and hence using it as the basis for adaptive hyperparameter scheduling, makes intuitive sense. While I have some minor qualms about the writing (see below), the paper was clear and easy-to-read overall. Lastly, the coverage of the huge amount of related work from the last 2-3 years is about as thorough as possible.

**Weaknesses:**

My biggests concerns with the paper are its limited novelty a lack of proper ablations.

Novelty:

None of the three mechanisms introduced by DrM are novel in-and-of themselves.
- As noted in the paper, "dormant-ratio-guided perturbation" follows the previous work of (D'Oro et al., 2023; Ash & Adams, 2020).
- Similarly, "dormant-ratio-guided exploitation" is based on the work of Ji et al. (2023).
- The idea of scheduling the exploration rate goes back a very long way, pre-dating deep RL. Some recent approaches, e.g., Agent57 (Badia et al.), also use a form of adaptive exploration.

What *is* new is the idea of using the dormant ratio to schedule the hyperparameters of these methods. I don't disagree that this is novel; however, it's rather limited novelty in my opinion.

Ablations:

While Figure 10 contains some ablations, the three mechanisms are each ablated in their entirety. I'd prefer it if only the adaptive scheduling component were ablated for each method, since this is the novel part. In other words, for a fair comparison, DrM ought to be compared against DrQ + perturbation resets (with fixed $\alpha$) + "Blended Exploration and Exploitation operator" (with fixed $\lambda$).

Moreover, the ablations only consider one domain (Adroit) and the results are aggregated across all Adroit tasks. I'd prefer to see more ablations in greater granularity. For me, they're the most important part of the experiments, since without them it's very hard to understand *why* the improvements work. (The argument that DrM lowers the dormant ratio and hence improves performance seems very "chicken-egg" to me; it's unclear which is the cause and which is the effect. Only the dormant-ratio-guided perturbations seem to directly target the dormant ratio.)

Minor things:
- Section 3.1 is too unscientific, e.g., the claim "This suggests that the dormant ratio acts as an intrinsic metric, influenced more by the diversity and relevance of the agent's behaviors than by its received rewards". This is all based on one example, and the claim that RL agents learn relevant behaviours before their returns improve sounds dubious to me. Another too-bold claim is: "As the dormant ratio captures the intrinsic characteristics of an agent's policy network, all the hyperparameters in this approach are robust to tasks and domains." This is never established, and in fact Table 1 in the Appendix suggests that some parameters of DrM require domain-specific tuning.
- The claim in the abstract that DrM has "no broken seeds (76 seeds in total)" is a little vague (what constitutes "broken"?) and it's never really returned to in the main body of the paper.

**Questions:**

- In the "dormant-ratio-guided exploitation" section, it's mentioned that "value underestimation occurs as the agent starts to acquire skills, given that $\pi$ is sub-optimal". I don't quite follow this. As the agent starts to improve, wouldn't we expect $\pi$ to become less sub-optimal?

---

> ### Author Response · Authors · 2023-11-19
> **Response to reviewer Jxw3 (1)**
>
> Thanks the Reviewer Jxw3 for your valuable review of our paper. We appreciate the questions you raised and are committed to delivering a comprehensive response to address the issues.
>
> > 1. Limited novelty as three mechanisms introduced in DrM closely follow the prior works.
>
> Our work does not claim to reinvent the three mechanisms from scratch. Instead, the central contribution of our study lies in the innovative application of the dormant ratio as a pivotal indicator of the agent's activity level. We uniquely employ the dormant ratio as a guiding principle, integrating it with the three pre-existing mechanisms. This integration allows for a dynamic balancing of exploration and exploitation strategies: the agent is directed to prioritize exploration when the dormant ratio is high, shifting its focus to exploitation as the dormant ratio decreases. We firmly believe that the novel application of the dormant ratio in orchestrating these mechanisms, supported by strong and extensive empirical results across a diverse range of challenging locomotion and manipulation tasks, adds significant value to the community of visual reinforcement learning.
>
>
>
> > 2. Ablations for adaptive scheduling components: compare DrM against DrQ with fixed factors for perturbation resets, blended exploration and exploitation operator.
>
>
> Thank you for your valuable suggestion regarding the ablation study. This is indeed a good baseline to compare with in order to show that the three dormant ratio-based components are crucial for the success of DrM. In response, we have updated Figure 10 to include the proposed comparison as an additional baseline. This revised figure now illustrates that DrM substantially outperforms DrQ-v2, with three added mechanisms being fixed and independent of the dormant ratio. This finding further strengthens the contribution of our algorithm, demonstrating its effectiveness and the significance of our approach to use the dormant ratio as a guiding principle.
>
>
> > 3. Ablations only consider one domain (Adroit) and the results are aggregated across all Adroit tasks.
>
> Although we believe that the results of three tasks from one domain are more than sufficient to demonstrate the effectiveness of each component, as opposed to many existing works that only conduct ablation study on one task, we include more ablation studies with 2 tasks on Meta-World to further substantiate our claim. We plot the results on 2 additional Metaworld tasks as well as 3 individual Adroit tasks (as opposed to the aggregated results of Figure 10) in Appendix G.
>
> > 4. The argument that DrM lowers the dormant ratio and hence improves performance seems very "chicken-egg". Only the dormant-ratio-guided perturbations seem to directly target the dormant ratio.
>
> We would like to emphasize, as illustrated in Figure 2 of the main text and Figure 11 in the Appendix, that the dormant ratio effectively indicates the agent's activity level. Notably, the reduction in dormant ratio consistently precedes any increase in reward. This observation suggests that it is not the increase in reward that causes the dormant ratio to drop. Instead, the decrease in dormant ratio leads the agent to actively explore its environment. Subsequently, as the agent begins to acquire reward signals, it starts to learn skills, resulting in an increase in rewards. This underscores the importance of the dormant ratio in the learning process of the agent.
>
>
> > 5a. Section 3.1 is too unscientific, e.g., the claim "This suggests that the dormant ratio acts as an intrinsic metric, influenced more by the diversity and relevance of the agent's behaviors than by its received rewards". This is all based on one example.
>
> We appreciate your concern regarding the basis of our claim. We would like to clarify that the claim is not solely based on a single example. In fact, the observation consistently emerges across our experiments spanning various tasks, and we have demonstrated this in Figure 11 of the original manuscript, which includes data from three additional DeepMind Control (DMC) tasks.
>
> To further address your concern and provide a clearer understanding, we provide additional qualitative visualizations in addition to the task Hopper Hop (Figure 1). These visualizations showcase the agent's behavior in relation to its dormant ratio throughout the training progress. We have made these visualizations available as videos on our anonymous project webpage, which can be accessed at https://drm-rl.github.io/asset/dormant.mp4.

---

> ### Author Response · Authors · 2023-11-19
> **Response to reviewer Jxw3 (2)**
>
> > 5b. The claim that RL agents learn relevant behaviours before their returns improve sounds dubious.
>
> In addition to the supplementary videos that qualitatively visualize the agent's behavior in relation to the dormant ratio, here we wish to highlight that when the dormant ratio is low, RL agents begin to execute meaningful actions, actively exploring their environment. This contrasts with scenarios where the dormant ratio is high, and the agent remains largely immobile. It's important to note that these "meaningful actions" are not always directly linked to the task specified by the reward function. Therefore, what we typically observe is a decrease in the agent's dormant ratio prior to any increase in reward. During this phase, although the task reward remains low, the agent begins to actively explore. Subsequently, this effective exploration leads to the agent learning task-relevant behaviors, culminating in an increase in task reward.
>
>
>
> > 6. Some parameters of DrM require domain-specific tuning, which is conflicted with the claim in the paper.
>
>
> Thank you for pointing this out. This indeed deserves further clarification. In Appendix D.2, we now present the performance of DrM across various tasks using a single, unified set of hyperparameters, as opposed to domain-specific ones. We observed that DrM off-the-shelf generally performs well across all tasks with this single set of hyperparameters. However, for certain tasks, finetuning specific DrM hyperparameters does yield additional performance improvements. It's also important to note that the hyperparameter tuning required for DrM is minimal. The adjustments we made were limited to the maximum perturbation rate, selecting from $0.6$ and $0.9$, and modifying the exploitation expectile value by selecting from $0.7$ and $0.9$.
>
>
>
>
> > 7. The claim in the abstract that DrM has "no broken seeds (76 seeds in total)" is a little vague (what constitutes "broken"?) and it's never really returned to in the main body of the paper.
>
> “Broken seeds” means seeds where the agents fail to learn a successful policy, and the episode reward of the agent remains low (close to zero) throughout the training process. We appreciate your suggestion on this point, and we have added the additional clarifications on broken seeds in the experiment section.
>
> > 8. In the "dormant-ratio-guided exploitation" section, it's mentioned that "value underestimation occurs as the agent starts to acquire skills, given that $\pi$ is sub-optimal". I don't quite follow this. As the agent starts to improve, wouldn't we expect $\pi$ to become less sub-optimal?
>
>
> Thanks for your insightful question. We have adjusted the writing of our manuscript to clarify this exploitation mechanism further. Here, the intuition is that, especially in the early stages of training, the replay buffer could contain a very small number of high-quality episodes (i.e., episodes with high reward) that the agent has encountered through exploration. However, in this training stage, $\pi$ is suboptimal, and so $Q$-value is often underestimated due to insufficient exploitation of high-quality samples in the replay buffer.
>
> Furthermore, this under-exploitation is for more than just the early stages of training. In hard visual continuous control tasks, what we have observed is that effective exploitation from high-quality examples is necessary for the agent to learn an optimal policy. This is evident from the ablation study of three Adroit tasks, where removing this mechanism results in significant performance degradation (Figure 9).
>
> Thus, to mitigate this underestimation, we introduce a value estimator $V$, which estimates the high expectile of $Q$, designed to converge more rapidly than $Q$-values. Therefore, this mechanism allows the RL agent to quickly exploit its historically successful trajectories without introducing additional overestimation.

---

> ### Comment · Reviewer_Jxw3 · 2023-11-21
>
> Firstly, thanks very much for the videos; I find them much more compelling as a motivational argument than the text in the paper, and I'd suggest emphasising the video links more in the paper (not just as a footnote). It's the sort of thing you really need to see to appreciate fully.
>
> Just a couple more questions if you don't mind:
> - In the videos shown, the background is basically static. Therefore, when the agent isn't moving, the whole screen is basically still. It occurs to me that this could be one reason why the dormant ratio grows large when the agent is inactive. (If the video background was "busier", perhaps it would cause more neurons to randomly light up.) Could you please comment on this? Do you think the heuristic would work as well in, say, videogames with prominent background animations? Could a simpler, alternative heuristic be: "increase the exploration rate when the screen is hardly changing?"
> - Edit: With regard to the above point, I'd particularly like to see a comparison against the baselines on the Cartpole *balance* task (not Cartpole *swingup*), since the objective here is to reduce movement.
> - Thankyou for including DrQ-v2 with Fixed Mechanisms; this is a very important comparison IMO. What were the values of the fixed hyperparameters that you used here?
>
> I'm basically happy with the rest of your responses. I agree with Reviewer jJ9Z's remark: "The connection between network activity and meaningful behaviors is still unclear, but I do understand that it is an empirical observation that they are correlated." I guess my main remaining concern is that this is just an empirical observation, and it's only really been established in continuous control tasks with static backgrounds. This could be a worthy contribution by itself, but I'd like to understand the limitations of the approach better.

---

> ### Author Response · Authors · 2023-11-22
>
> 1. Thank you for your insightful question. In Appendix K, we have conducted three additional experiments, including the one you requested for Cartpole Balance, to demonstrate that DrM is not simply maximizing the velocity of changes between frames to lower the dormant ratio and encourage exploration.
>
>     First, we show that simply adding exogenous noise into the background does not change our insight of the dormant ratio. We performed an experiment using the DMC-Generalization Benchmark, where, unlike standard DMC tasks, DMC-Generalization introduces a dynamic element into the background by inserting a video clip. Contrary to the hypothesis that a low dormant ratio is tied to major frame-to-frame changes, our findings in DMC-Gen do not support this. The dormant ratio pattern in DMC-Gen is similar to that in regular DMC tasks, where the agent's dormant ratio starts high and then drops abruptly at a certain point during training. This challenges the idea that the dormant ratio is solely a measure of frame-to-frame change velocity.
>
>
>     Next, if DrM is simply maximizing the difference between frames, then a simple intrinsic reward approach to maximize the difference between consecutive frames should achieve a comparable performance. To test this, we implemented a baseline experiment on three Adroit tasks, where the intrinsic reward is the L1 difference between the first and last frames in the agent's observation stack (In DrM, we follow the standard that the agent's observation at each timestep is a stack of three consecutive image frames). Preliminary results (3M frames) indicate that simply maximizing frame differences is not effective, and we will update with the final results upon completion.
>
>     Finally, as requested, we have tested DrM, along with other baseline algorithms, on the Cartpole Balance Sparse task. Although this task is simple, DrM still outperformed the baselines. Moreover, in tasks such as Acrobot Swingup Sparse, Humanoid Stand, and Dog Stand, where excessive motion could result in low rewards, DrM's impressive performance further illustrates that reducing the dormant ratio is not just about maximizing frame changes.
>
>     Thank you again for your question. We believe these three additional experiments adequately address your concerns. Should you have any further questions, we are more than willing to provide further clarification.
>
>
>
> 2. Regarding the hyperparameter, we use a constant perturbation factor 0.9, we set the expectile value $\tau$ to be $0.7$, and for the fixed exploitation hyperparameter $\lambda$, we notice that it is the most sensitive hyperparameter based on the observation of Ji et al. So we search over the best $\lambda$ from $0.4$,$0.5$, or $0.6$ from the Pen task and then test the performance on the three Adroit tasks altogether.

---

> > ### Comment · Reviewer_Jxw3 · 2023-11-22
> >
> > Thanks very much for your responsiveness and the new experiments. Regarding Figure 20, I wasn't so much thinking of using the difference in frames as an intrinsic reward, but rather using the running average to schedule the hyperparameters of the three mechanisms, similar to how the dormant ratio is used. Anyway, I wasn't clear about this, and I'm satisfied that the additional experiments rule out my hypothesis. I'd still love to better understand why the approach works so well, but there's enough contribution here as it is for acceptance. I've raised my score accordingly.

---

> > > ### Author Response · Authors · 2023-11-23
> > > **Response to Reviewer Jxw3**
> > >
> > > We appreciate your recognition of our contribution and additional experiments. We are pleased that we could address your concerns. Thank you for the effort you put forth and the valuable insights you provided during the review and rebuttal periods.

---

### Official Review · Reviewer_R5aS · 2023-10-30

**Soundness:** 3 good
**Presentation:** 3 good
**Contribution:** 3 good
**Rating:** 6
**Confidence:** 3

**Summary:**

This paper presents an algorithm which leverages the fraction of dormant units in the network to guide exploratory behaviour. The resulting method demonstrates significant improvements over baseline algorithms that use naive exploration strategies on a variety of challenging robotic manipulation domains.

**Strengths:**

- The paper evaluates on more than one domain
- The proposed method significantly outperforms the baselines compared against.
- The paper performs an ablation on a single environment of the different components of the proposed method
- Nice contrast with other adaptive exploration methods which use e.g. bandit methods to select epsilon-greedy exploration parameter.
- Pseudocode is provided for each component of the proposed method
- The proposed method depends on a single easy to measure property of the network state
- The correlation between the dormant ratio and the qualitative properties of the network's exploration is intriguing.
- The idea of strategically exploring based on properties of an agent's representation is a new and exciting approach which I think could be applied more broadly in RL and could be really useful to the community.

**Weaknesses:**

- The design choices going into the main method proposed by the paper are not given sufficient motivation, and I am concerned that although the algorith does indeed improve performance, it does not do so for the reasons claimed. In particular:
        - Why is the method using shrink and perturb rather than directly resetting inactive units? Is it possible that S&P is introducing additional exploration due to the noisy updates to the parameters, similarly to the "noisy nets" strategy of Fortunato et al.?
        - The exploration strategy appears to actually introduce *more* noise after the network is "activated" than it used before activation due to the max operator. However, the motivation for this noise adaptation is to introduce more noise early in training when the network's behaviours are less interesting. I agree that this should have the effect of increasing exploratory behaviour throughout training compared to using a schedule, but not in the way claimed by the paper.
        - What motivates the modelling assumption that the network only becomes "activated" once in its training trajectory? Why is it assumed that the number of inactive units will not drop repeatedly?
        - The use of expectiles introduces a large benefit in training based on the ablation study, but the optimal expectile seems to depend on the environment. Is there additional justification for using expectiles beyond the empirical benefit and the additional knob to tune in the training algorithm?
- I have some concerns about the fairness of the comparisons employed in this work:
        - Each of the components of the method seem to increase exploratory behaviour. However, the baselines compared against do not incorporate any particularly sophisticated exploration strategies (DrQv2 uses a linear scheduler and it doesn't look like A-LIX or TACO improve upon this).
        - The proposed algorithm has a number of hyperparameters which have different optimal values for different environments. As a result, I'm not sure how much of the performance gain observed in the paper is due to finetuning of these additional hyperparameters vs the fundamental benefits of the method.
        - Given the different optimal values of the expectile and perturbation scale for different environments, the robustness of the method to its hyperparameters is a major concern to me. How much worse is performance if the optimal hyperparameters for one domain are used in another?
- While I appreciate the pseudocode provided in the appendix, there are some issues with clarity. For example, algorithm 2 says that one creates a deep copy of the network and stores this in the variable new_net, but then the next line randomly initializes the weights of new_net. What did the copy step do? Algorithm 1 lines 6-7 are also extremely vague and should be clarified with an equation.
- Related to my concerns about design choices, there are many aspects of the algorithm that seem unnecessarily complicated -- for example, one could simply use an exploration noise value equal to the sigmoid of the dormant unit fraction discussed, or use a maximal exploration noise until the awakened threshold is met and then start the noise decay. I would appreciate ablations which illustrate the importance of these design choices.
- The use of the term "exploitation" seems inaccurate to me. Exploitation is usually used to refer to greedy behaviour with respect to the network's predictions. In this case, exploitation refers more to optimistic Q-target updates which bias towards higher expectiles, encouraging the network to visit states which have occasionally yielded high rewards in the past. This approach would presumably lead to overestimation of Q-values in noisy environments and potentially result in suboptimal policies where the agent is attracted to high-variance, low-expectation states. I think the paper would benefit from a clearer discussion of what precisely this component of the method is doing, and potentially also a rebrand to characterize it more as a "risk-seeking" parameter rather than an "exploitation" parameter.

**Questions:**

- There is a lack of clarity in the causality of the phenomenon studied by the paper: is the claim that interesting behaviour leads to greater data variety and thus less overfitting and inactive units, or is the idea that reduced representation capacity limits the range of behaviours that a network can express?
- How sensitive is the method to hyperparameters?
- Could the authors provide additional justification for the expectile prediction component of their method, and illustrate what types of environments this might prove detrimental for?
- Can the authors comment on the design choices of their algorithm highlighted in the "weaknesses" section and provide justification for why the variant used in the paper is employed, rather than a simpler and more direct version of achieving the goal claimed for each component? For example, why the method perturbs all units rather than resetting dead ones, why exploration noise is potentially greater later in training than earlier due to the max operator, and why the expectile method which is described as *exploiting* past success should have the effect of prioritizing high-variance states and thus correspond to greater *exploratory* behaviour?
- How robust is the algorithm to the trajectory of dead units in networks trained on a particular environment? For example, if we considered a different architecture or set of tasks (perhaps ProcGen or Atari), would the strategies employed by the proposed method still provide the same benefit, or would they need to be re-tuned?

---

> ### Author Response · Authors · 2023-11-19
> **Response to reviewer R5aS (1)**
>
> Thanks Reviewer R5aS for the valuable review of our paper. We appreciate the questions you raised and are committed to delivering a comprehensive response to address the issues.
>
>
> **Design choices of DrM**
>
>
> 1. **Dormant Ratio Based Perturbation**
> Dormant ratio based perturbation is a simple mechanism that periodically perturbs the network by scaling down the original network weight (with magnitude of the scaling/perturbing controlled by dormant ratio) and injecting some noise into it.
>
> > Q1: Why is the method using shrink and perturb rather than directly resetting inactive units?
>
> Thanks for your thoughtful question. We conducted an experiment comparing our approach with directly resetting the inactive (dormant) neurons in the updated Appendix F. The results clearly demonstrate that our method significantly outperforms the approach of directly resetting the weights of dormant neurons. Upon analyzing the results, simply resetting weights of dormant neurons does not consider the holistic effect on the entire layer of the network. For example, a part of neurons in the middle layer cannot directly determine the output in a specific action dimension. In a fully connected network, we need to contemplate resetting all neurons rather than selectively discarding weights of dormant neurons. Additionally, neurons that are dormant but have large gradients during a certain period might become active in the next period, making it challenging to determine which neurons' weights should be retained or reset based on a single batch of data. In contrast, our perturbation-based approach encourages exploration, effectively reducing the dormant ratio, enabling the agent to learn better behavior, and ultimately achieving better performance.
>
>
> > Q2: Is it possible that S&P is introducing additional exploration due to the noisy updates to the parameters, similarly to the "noisy nets" strategy of Fortunato et al.?
>
> Thank you for your insightful question. We would like to clarify that the perturbation mechanism in DrM is fundamentally different from the NoisyNet technique mentioned. In NoisyNet, exploration is encouraged by perturbing the output of the policy/value network, achieved by injecting noise into the weights of the neural network at every timestep. In contrast, our approach in DrM involves updating or refreshing the weights only after a relatively long fixed time interval (2e+5 frames). This key difference underlines the distinct nature of our perturbation mechanism compared to the NoisyNet strategy. Nevertheless, we acknowledge the relevance of the NoisyNet strategy to our work and will include it in our reference list. We have also updated our writing in Section 3.2 accordingly to emphasize the distinction from NoisyNet.
>
>
> 2. **Awaken Exploration Scheduler**
>     > Q1: One could simply use an exploration noise value equal to the sigmoid of the dormant unit fraction discussed, or use a maximal exploration noise until the awakened threshold is met and then start the noise decay
>
>     Our objective here was just to introduce a straightforward modification to the existing linear exploration schedule in DrQ-v2, aiming to provide the agent with higher exploration noise when the dormant ratio is high. Although the equation $\displaystyle \frac{1}{1+exp(-(\beta-\hat{\beta})/T)}$ may seem complex at first glance, it essentially aligns with your suggestion of "using maximal exploration noise until awakening." The key difference is only the adaptive adjustment of exploration noise based on the dormant ratio rather than employing a fixed uniform value, as you propose. This nuanced adjustment allows for a more dynamic and responsive exploration strategy tailored to the evolving state of the dormant units. While the key insight here is that dormant ratio can be leveraged as a key tool for efficient exploration in visual RL, our method does not exclude some modifications of our design choice, and we welcome future works to build on top of DrM to perhaps come up with better exploration techniques.
>
>
>
>     > Q2: The exploration strategy appears to actually introduce more noise after the network is "activated" than it used before activation due to the max operator.
>
>     Thanks for your insightful question. This is actually not the case empirically. The incorporation of the max operator in our method serves primarily to prevent a substantial rise in the dormant ratio after initial activation. In practice, however, this operator is almost never invoked. Once the dormant ratio is awakened, it usually decreases sharply, making the second term, $\sigma_\text{linear}(t − t_0)$, the predominant factor. Consequently, in Equation 3, keeping only $\sigma_\text{linear}(t − t_0)$ post-awakening would not significantly change the exploration schedule.

---

> > ### Comment · Reviewer_R5aS · 2023-11-19
> >
> > Thanks to the authors for their detailed response. I have a few follow-up questions:
> >
> > “In contrast, our perturbation-based approach encourages exploration, effectively reducing the dormant ratio, enabling the agent to learn better behavior, and ultimately achieving better performance.”
> > - Thanks for adding the evaluation of ReDo in Appendix F. I see that the ReDo agent does not include the adaptive exploration scaling or the expectable regression components of the method, which makes it difficult to disentangle the effect of perturbation vs resetting dormant neurons. A more informative comparison would be ReDo + awaken + expectiles vs S&P + awaken + expectiles.
> >
> > “The key difference is only the adaptive adjustment of exploration noise based on the dormant ratio rather than employing a fixed uniform value, as you propose. This nuanced adjustment allows for a more dynamic and responsive exploration strategy tailored to the evolving state of the dormant units."
> > -  I appreciate that this could in theory allow for annealed exploration as the dormant ratio declines, but I think evidence showing the importance of this property is missing from the paper. Did the authors ever evaluate a baseline which uses a value of 1 prior to awakening?

---

> ### Author Response · Authors · 2023-11-19
> **Response to reviewer R5aS (2)**
>
> 3. **Dormant-ratio-guided exploitation:**
>     > Q1. The use of the term "exploitation" seems inaccurate to me.
>
>     > Q2. Could the authors provide additional justification for the expectile prediction component of their method, and illustrate what types of environments this might prove detrimental for?
>
>
>
>     Thank you for raising this important point. We acknowledge that our use of the term "exploitation" might require further clarification. In our context, "exploitation" refers to a strategy that could allow the RL agent to quickly exploit its serendipitous, unexpectedly successful experiences from its exploration process. The key insight that we observe is that, especially in the early stages of training, the replay buffer could contain scarce high-quality episodes (i.e., episodes with high reward) that the agent has encountered through exploration. In this training stage, $\pi$ is suboptimal, and the $Q$-value is often underestimated due to insufficient exploitation of high-quality samples in the replay buffer. Thus, to mitigate this underestimation, we introduce a value estimator $V$, which estimates the high expectile of $Q$, designed to converge more rapidly than $Q$-values. Therefore, this mechanism allows the RL agent to quickly exploit its historically successful trajectories **without introducing additional overestimation**. (This is systematically shown in Ji et al.)
>
>
> 4. **Dormant Ratio and Exploratory behaviors:**
>
> > There is a lack of clarity in the causality of the phenomenon studied by the paper: is the claim that interesting behaviour leads to greater data variety and thus less overfitting and inactive units, or is the idea that reduced representation capacity limits the range of behaviours that a network can express?
>
> The motivation behind our work stems from the observation that the policy-centric data distribution often leads to overfitting, resulting in the network losing its capacity to learn more complex behaviors. This situation causes the network to become inactive, and the inactivity further restricts policy exploration. Therefore, we employ the method of perturbing the network and adjusting exploration strategies based on the dormant ratio. The goal is to revive the network's learning capabilities, promoting the emergence of diverse behaviors that are essential for effective learning and decision-making in RL environments.

---

> > ### Comment · Reviewer_R5aS · 2023-11-20
> >
> > 3. Thanks to the authors for clarifying the annealing of the expectiles. In theory overestimation might still be an issue in tasks where networks accumulate and maintain nontrivial numbers of dormant neurons, but I agree that this doesn't occur in the domains studied in this paper; it could also presumably be resolved by using a similar max over the exponential and a linear decay term as is done in the 'awaken' scheduler, so doesn't seem to be as big of a problem as I'd originally interpreted it to be.
> >
> > 4. I agree with the authors that the issue of exploration and representational capacity is a bit circular, but I think it would be informative to find a way of disentangling the effect the method is having on each of these factors: in particular, I can imagine that the effect of shrink-and-perturb is two-fold: first, it resets the network parameters enough to remove dormant neurons, and second it introduces a form of "deep exploration", akin to the ensembling strategy used in bootstrapped DQN but applied to sequential random perturbations of the parameters rather than to an ensemble. I would hypothesize that the increase in "dithering" exploration introduced by the awaken scheduler and the deeper exploration induced by network resets are complementary to each other, and it would be nice to find a way of testing whether this is the case -- for example, if the random perturbation is always equal to the original initialization parameters, how much does this reduce the efficacy of S&P?

---

> ### Author Response · Authors · 2023-11-19
> **Response to reviewer R5aS (3)**
>
> 5. **Hyperparameters**
>
> > a. The proposed algorithm has a number of hyperparameters which have different optimal values for different environments. As a result, I'm not sure how much of the performance gain observed in the paper is due to finetuning of these additional hyperparameters vs the fundamental benefits of the method.
>
> > b. The use of expectiles introduces a large benefit in training based on the ablation study, but the optimal expectile seems to depend on the environment. Is there additional justification for using expectiles beyond the empirical benefit and the additional knob to tune in the training algorithm?
>
> > c. Given the different optimal values of the expectile and perturbation scale for different environments, the robustness of the method to its hyperparameters is a major concern to me. How much worse is performance if the optimal hyperparameters for one domain are used in another?
>
> Thank you for pointing this out. This indeed deserves further clarification. In Appendix D.2, we now present the performance of DrM across various tasks using a single, unified set of hyperparameters, as opposed to domain-specific ones. We observed that DrM off-the-shelf generally performs well across all tasks with this single set of hyperparameters. However, for certain tasks, finetuning specific DrM hyperparameters does yield additional performance improvements. It's also important to note that the hyperparameter tuning required for DrM is minimal. The adjustments we made were limited to the maximum perturbation rate, selecting from $0.6$ and $0.9$, and selecting the exploitation expectile value from either $0.7$ or $0.9$.
>
>
>
>
> 6. **Compare with baselines with carefully designed exploration strategies**:
>
> > Q: I have some concerns about the fairness of the comparisons employed in this work: Each of the components of the method seem to increase exploratory behaviour. However, the baselines compared against do not incorporate any particularly sophisticated exploration strategies (DrQv2 uses a linear scheduler and it doesn't look like A-LIX or TACO improve upon this).
>
> Thank you for your valuable feedback. In response, we have added a new section in the appendix (Appendix E), where we conduct additional experiments on three Adroit tasks to compare DrM against an intrinsic exploration baseline.
>
> Specifically, we compare DrM with Random Network Distillation (RND), a well-known and extensively used method in intrinsic reward-based exploration. We integrated RND into the existing DrQ-v2 implementation to evaluate its performance. The results, detailed in Appendix E and Figure 8, reveal that while RND enhances DrQ-v2 through intrinsic exploration, DrM still markedly outperforms the RND baseline in three Adroit tasks.
>
>
>
>
> 7. **Explanation of details in pseudocode**
>
> > Q: While I appreciate the pseudocode provided in the appendix, there are some issues with clarity. For example, algorithm 2 says that one creates a deep copy of the network and stores this in the variable new_net, but then the next line randomly initializes the weights of new_net. What did the copy step do? Algorithm 1 lines 6-7 are also extremely vague and should be clarified with an equation.
>
> Regarding Algorithm 2, we acknowledge that the step of creating a deep copy of the network into new_net followed by the random initialization of its weights was redundant. The intention was to replicate the network's architecture, not its weights. We have since revised this section in the paper to clarify that only the network's shape is copied.
> For Algorithm 1, lines 6-7, we appreciate your input on the vagueness of the description. We have updated the paper to include a more detailed explanation to clearly articulate the calculation process.

---

> ### Author Response · Authors · 2023-11-19
> **Response to reviewer R5aS (4)**
>
> 8. **Other Questions**:
>
> > 1. What motivates the modelling assumption that the network only becomes "activated" once in its training trajectory? Why is it assumed that the number of inactive units will not drop repeatedly?
>
> This is primarily based on extensive empirical observations from our experiments. We have found that once a network is activated and begins to explore meaningful behaviors or strategies, it is unlikely to revert to a state of inactivity. This tendency towards sustained activation, post-initial exploration, is a pattern we've consistently observed, suggesting that the network maintains a level of engagement with the task at hand after its initial activation phase.
>
> > 2. How robust is the algorithm to the trajectory of dead units in networks trained on a particular environment? For example, if we considered a different architecture or set of tasks (perhaps ProcGen or Atari), would the strategies employed by the proposed method still provide the same benefit, or would they need to be re-tuned?
>
> DrM is primarily designed for continuous control tasks, which arguably holds more practical value compared to environments like ProcGen or Atari games since it is closely related to real-world robotics applications. In the realm of continuous control, DrM has been proven effective across 19 of the most challenging visual RL tasks spread over three distinct domains, from locomotion to robotic manipulation. We believe that the diversity of the tasks conducted in DrM, as presented in our paper, adequately demonstrates DrM's broad applicability and robustness.

---

> > ### Comment · Reviewer_R5aS · 2023-11-20
> >
> > "DrM is primarily designed for continuous control tasks": I agree that the diversity of environments within the continuous control/robotic simulation domains supports the claim that this is an effective method for continuous control tasks. Can the authors comment on what aspects of the algorithm they think will/won't generalize to other domains? This would be a helpful discussion to include in the paper as well.

---

> > > ### Author Response · Authors · 2023-11-21
> > > **Response to Reviewer R5aS**
> > >
> > > > A more informative comparison would be ReDo + awaken + expectiles vs S&P + awaken + expectiles.
> > >
> > > Thank you for your advice. We have carried out further experiments and detailed the findings in Appendix F. As depicted in Figure 15, our method notably surpasses both the strategy of resetting solely dormant neurons and the DrM with ReDo approach. We speculate that this improvement is due to the positive impact of resetting non-dormant neurons on exploration. Additionally, our use of the dormant ratio to guide exploration strategy distinguishes our approach from previous works.
> > >
> > > > I appreciate that this could in theory allow for annealed exploration as the dormant ratio declines, but I think evidence showing the importance of this property is missing from the paper. Did the authors ever evaluate a baseline which uses a value of 1 prior to awakening?
> > >
> > > To answer your question, we have conducted additional experiments on Adroit, where we set the exploration schedule exactly the same as you asked here: maximal exploration noise (1) before awakening and then using the linear schedule afterward. The results are updated in Appendix I. We find that using the maximal exploration noise instead of our adaptive adjustment of exploration noise based on the dormant ratio results in significant performance degradation across all three tasks. These added experimental results should justify our design choice.
> > >
> > > > If the random perturbation is always equal to the original initialization parameters, how much does this reduce the efficacy of S&P?
> > >
> > > Thank you for your insightful observation and hypothesis. We conducted experiments by replacing reinitialized perturbations with the original initialization parameters to assess the impact on the efficacy of S&P on Sweep-Into and Stick-Pull. The results indicate that 1-shrink perturbations caused only a 10% decrease in performance in Sweep-Into, while maintaining comparable performance in Stick-Pull. These findings suggest that the increase in "dithering" exploration introduced by the awaken scheduler and the deeper exploration induced by network resets may indeed be complementary. We appreciate your suggestion, and these experiments provide valuable insights into the interplay between different exploration strategies. If you have any further questions or if there are specific aspects you'd like us to delve into, please feel free to let us know.
> > >
> > > > Can the authors comment on what aspects of the algorithm they think will/won't generalize to other domains?
> > >
> > > In this work, we primarily concentrate on continuous control and actor-critic-based algorithms, which dominate the existing works for (visual) continuous control. However, while some specific design choices may need modification for applications using DQN/Efficient Rainbow instead of DrQ-v2, the three key mechanisms we propose for DrM could still be adapted for discrete action tasks with some minor adjustments. For instance, the dormant-ratio-based exploration noise schedule could be replaced by a dormant-ratio-based $\epsilon$-greedy strategy. The shrink and perturb and dormant-ratio-guided exploitation mechanisms should also be applicable to DQN, albeit with adjustments to their hyperparameters.
> > > While our current work focuses on visual motor control, we welcome future research to adapt and apply our methodologies to discrete action spaces.
> > >
> > > We sincerely appreciate your invaluable suggestions and look forward to further discussions and considerations of your score before the conclusion of the discussion period.

---

> > > > ### Comment · Reviewer_R5aS · 2023-11-21
> > > >
> > > > Thanks for the quick turnaround on the additional ablations! The added results have convinced me that the paper's strategy of having each component depend on the dormant neuron ratio is critical to its success and can't be simplified by the strategies I suggested. I would have liked to see a more fine-grained explanation of _why_ these components are so critical, but I don't think this should necessarily be a barrier to acceptance. Additionally, the findings on the relative robustness of the method to a fixed set of hyperparameters has mollified my concerns about the dependence of the method on its hyperparameters. The only remaining worry I have is whether the correlation between the dormant neuron ratio and behavioural complexity will hold across more diverse sets of domains; however, I think even if it turns out that the method is benefiting from a particular correlation between learning progress and dormant ratios that only holds in the types of simulated domains studied in the paper, this is still a sufficiently general and widely-used set of benchmarks that the finding is interesting on its own. I have updated my score accordingly.

---

> > > > > ### Author Response · Authors · 2023-11-23
> > > > > **Response to Reviewer R5aS**
> > > > >
> > > > > We are glad to see that our response addressed your concerns, and appreciate your acknowledgment for our paper. Thank you for the patient time and valuable suggestions you invested in the review and discussion.

---

### Official Review · Reviewer_2E41 · 2023-10-31

**Soundness:** 3 good
**Presentation:** 3 good
**Contribution:** 3 good
**Rating:** 8
**Confidence:** 4

**Summary:**

The paper provides insight that the dormant ratio of the network acts can indicate if the agent is effectively exploring or not. Based on this insight, several mechanisms are proposed to use the dormant ratio to handle the exploration-exploitation trade-off. The results show that the proposed mechanism significantly improves performance and sample efficiency over existing methods.

**Strengths:**

The central insight provided in the paper is very interesting. The main idea is that when the dormant ratio is very high, the agent is not getting diverse experience, so it would be good to increase exploration at this point. The insight is verified by looking at the dormant ratio and the agent's behaviour for some seeds.
This insight is then used to control the exploration-exploitation trade-off. Using an internal indicator like the dormant ratio is a promising direction to tackle the exploration-exploitation dilemma as it does not depend on any external signals like reward.

The results show that the proposed method, DrM, is effective in many visual RM domains and significantly outperforms existing methods.

**Weaknesses:**

Although the paper has interesting ideas and promising results, some of the weaknesses stop me from recommending a full acceptance of the paper.
-**Weak empirical evaluation** All of the experiments are performed with just four random seeds, which raises questions about the statistical significance of the results. I refer the authors to Patterson et al. (2023) on how to perform good empirical experiments. I realize that it might not be feasible to do 30 runs for all environment-algorithm pairs, but there should be at least one experiment in the paper that will stand the test of time. I suggest that the authors perform at least ten runs for all algorithms in some environments, maybe on the four dense tasks in MetaWorld (from Figure 7).
-**Too strong claims** For example, the last line of Section 3.1 says, " ... all the hyperparameters in this approach are robust to tasks and domains.". However, no evidence is provided for this claim. Their method introduces 4 or 5 new hyper-parameters, and different values are used for different environments. This means that the same hyperparameter is not optimal across domains.

Patterson, A., Neumann, S., White, M., & White, A. (2023). Empirical Design in Reinforcement Learning. arXiv preprint arXiv:2304.01315.

**Questions:**

What value of $\hat{\beta}$ is used in Figures 3 and 4? This should be specified in the figure itself.

What exactly is the linear schedule, $\sigma_{linear}$, of exploration? The paper says that it's the same as defined in DrQ-v2. However, I believe the paper should be as self-contained as possible, so details like this should be provided in the appendix.

---------------

I've increased my score as new results largely overcome my initial concerns.

---

> ### Author Response · Authors · 2023-11-19
> **Response to reviewer 2E41**
>
> Thanks Reviewer 2E41 for the valuable review of our paper. We appreciate the questions you raised and are committed to delivering a comprehensive response to address the issues.
>
> 1. Weak empirical evaluation
>
> > All of the experiments are performed with just four random seeds, which raises questions about the statistical significance of the results. ... I suggest that the authors perform at least ten runs for all algorithms in some environments, maybe on the four dense tasks in MetaWorld (from Figure 7).
>
> Thank you for raising the question of the limited number of random seeds in our experiments and its possible effect on statistical significance, as referenced in Patterson et al. (2023). It is indeed crucial to increase the number of random seeds for some algorithms that show only marginal improvements. However, in our case, we wish to emphasize that DrM exhibits a substantial performance boost compared to other existing algorithms. The variation of performance across random seeds cannot be the reason to support the consistent and huge performance gain of DrM against all baseline algorithms and across all tasks.
> Nevertheless, understanding the value of robust empirical evidence, we conducted 10 runs of experiments for both DrQ-v2 and DrM across three MetaWorld environments in Appendix H, and cited Patterson et al. (2023) in our reference list. It is evident that DrM is not sensitive to the randomness of seeds, and it consistently maintains a significant performance lead over baseline algorithms.
>
>
> 2. Too strong claims
>
> >For example, the last line of Section 3.1 says, " ... all the hyperparameters in this approach are robust to tasks and domains.". However, no evidence is provided for this claim. Their method introduces 4 or 5 new hyper-parameters, and different values are used for different environments. This means that the same hyperparameter is not optimal across domains.
>
> Thank you for your feedback. We have updated our Appendix D accordingly. We would like to point out that the hyperparameter tuning required for DrM is minimal. The adjustments we made were limited to two hyperparameters: the maximum perturbation rate, selecting from $0.6$ and $0.9$, and modifying the exploitation expectile value by selecting from $0.7$ and $0.9$.
> More importantly, in Appendix D.2, we now present the performance of DrM across various tasks using **a single, unified set of hyperparameters**, as opposed to domain-specific ones. We observed that DrM off-the-shelf generally performs well across all tasks with this single set of hyperparameters. However, for certain tasks, finetuning specific DrM hyperparameters does yield additional performance improvements.
>
>
> > 3. What value of β is used in Figures 3 and 4? This should be specified in the figure itself.
>
> Thank you for your feedback. In those figures, $\beta=0.2$. We have updated the results in the paper.
>
> > 4. What exactly is the linear schedule of exploration? The paper says that it's the same as defined in DrQ-v2. However, I believe the paper should be as self-contained as possible, so details like this should be provided in the appendix.
>
> Thanks for the valuable suggestion. We have included this in the Table of Appendix D.1. The linear schedule stands for linearly annealing the std of exploration noise until a given number of exploration frames.

---

> > ### Comment · Reviewer_2E41 · 2023-11-23
> >
> > Dear Authors, Thank you for adding additional results. These new results largely overcome my initial concerns. And in light of these new results, I have updated my score. However, I would push back against the claim that "DrM exhibits a substantial performance boost compared to other existing algorithms."  In Metaworld environments in Figure 7, the difference between TACO and DrM is not statistically significant. Performing more runs is the only way to conclude which algorithm performs better.

---

> > > ### Author Response · Authors · 2023-11-23
> > > **Response to Reviewer 2E41**
> > >
> > > We are grateful for your acknowledgment of our responses and the adjusted score. Concerning your suggestion regarding Figure 7, we will refine our description of MetaWorld performance. Following the rebuttal period, we plan to augment the number of training seeds on MetaWorld to more effectively highlight DrM's advantages. Your insights are crucial for elevating the overall quality of our paper.

---

### Official Review · Reviewer_jJ9Z · 2023-11-01

**Soundness:** 3 good
**Presentation:** 3 good
**Contribution:** 4 excellent
**Rating:** 6
**Confidence:** 4

**Summary:**

This paper identifies a pervasive issue in pixel-based RL, where the agent's
policy has few activations and limited activity in the early phases of learning.
Related to this phenomenon is neuron dormancy, and the authors propose an
algorithm that uses this insight in three mechanisms: weight perturbation,
exploration and exploitation. The combination of these three improvements is
referred to as Dormant ratio Minimization (DrM), and the authors claim that this
algorithm is the first (documented and model-free) algorithm to solve dog and
manipulator, demonstration-free adroit, and is generally sample efficient in
various environments.

**Strengths:**

- The insight is derived from seemingly unrelated but recent work on Neuron
  Dormancy, which explains how the phenomenon can impact learning in neural
  networks through a potential plasticity mechanism. This paper seems to
  provides a novel lens for the phenomenon, showing that neural activity can be
  correlated with exploration, and that explicitly minimizing dormancy,
  reinforcement algorithms can be improved in their ability to both explore and
  'exploit'.
- The paper is overall well written. The empirical analysis is good and,
  although the statistical power is low due to a small sample size, the
  improvements seem statistically significant. I particularly like that the
  dormancy ratio is in-fact minimized, because the algorithm seems to be doing
  this implicitly. The algorithm does seem to achieve SOTA on a few challenging
  continuous control problems with relevant baselines. Overall, the
  contributions are convincing.

**Weaknesses:**

- The foundational motivation for dormancy ratio minimization is not entirely
  convincing. It is not clear why high neuron dormancy should necessarily lead
  to a decrease in exploration, even if this was empirically observed. It is
  also not that clear why a lower neuron dormancy may lead to more exploration.
  I think the paper would benefit from a more careful treatment of this finding,
  either with further experiment, a toy model/study, theory or merely some text
  describing why this should be expected.
- There are a few aspects of the proposed algorithm that are mildly worrying.
  First is the exploitation mechanism. I do not understand how exploitation is
  determined by a hyperparameter controlling a value update towards a
  state-value or an action-value.
- The second thing is that the algorithm
  proposes three mechansism, each with a hyperparmeter that governs a schedule.
  This provides several degrees of freedom to improve upon the base algorithm,
  and no hyperparameter study is conducted. This is not good empirical science,
  but the empirical demonstration of success on hard problems does at least hold
  promise for further work.

Overall, I think the first and second weakness are both more pressing and addressable within a rebuttal period.

**Questions:**

- Section 3 (Language surrounding Dormant Ratio, specifically beginning of 3.2): While you demonstrate the empirial obervation that low neuron dormancy is correlated with meanginful high-level behavior, it is overclaiming to say that it is essential. I can imagine an approach to construct a neural network with an arbitrarily high dormancy ratio by encapsulating any policy within a much larger but inactive network. Thus, it is not necessarily the case that high dormancy ratio translates to less meaningful behavior, but that it can be correlated with it during training.
- Section 3 (Expectile Regression and Exploitation): This section seems to build off recent work, and I am not familiar with "blended exploitation and exploration". However, I do not see how the proposed method has anything to do with exploitation. Exploitation and exploration are fundamentally about control and action, but this section is primarily about the target of a temporal difference method. But even putting that aside, I do not see why placing higher weight on the state value function should empahsize exploitation, or why placing higher weight on the action value function should emphasize exploration.
- Section 3 (Hyperparameters): The method involves three components, each with at least one hyperparameter. This gives your proposed algorithm a number of additional degrees of freedom over the base algorithm and the other baselines. Looking at the experiments, there is an ablation study for each individual component but no results on the sensitivity to the various hyperparameters. It would have been good to explore this, at least in one of the simpler environments.
- Section 4 (Dormant ratio analysis): Validating the fact that the dormancy ratio is indeed minimized vs the base algorithm is interesting. One thing that would strengthen these results further is to show that the other baselines (A-LIX, TACO) are similar to DrQ-v2 in that they do not minimize the dormancy ratio on at least some problem. It would also be interesting to investigate why there is some periodicity in the dormancy ratio such as in humanoid run and manipulator.
- Section 4 (Ablation study): I appreciate the effort to ablate your algorithm in Adroit, a relatively complex environment. One thing that would help is to show whether each component of DrM is faithful to its motivation. For example, the exploration mechanism could be shown to have a large influence on performance in environments that require exploration and less influence on environments that do not require as much exploration. This would also serve as an opportunity to further elaborate on what the exploitation mechanism accomplishes.

# Minor
- Section 2 (visual RL): Is it necessarily the case that your setting is POMDP and that the observations provided to the agent lack some information? While this can be true in some settings, the environment is usually constructed so that the observatiosn do include all necessary information (e.g. concatenation of frames in Atari).
- Section 2 (Dormant Ratio): You use the term "Linear layer" in both definitions, but I am not sure whether this is required in the definition or what it is that you mean. Does it mean the last layer that usually maps the penultimate activations to the output?

---

> ### Author Response · Authors · 2023-11-19
> **Response to reviewer jJ9Z (1)**
>
> Thanks Reviewer jJ9Z for the valuable review of our paper. We appreciate the questions you raised and are committed to delivering a comprehensive response to address the issues.
>
> > 1.  The foundational motivation for dormancy ratio minimization is not entirely convincing. It is not clear why high neuron dormancy should necessarily lead to a decrease in exploration, even if this was empirically observed. It is also not that clear why a lower neuron dormancy may lead to more exploration. I think the paper would benefit from a more careful treatment of this finding, either with further experiment, a toy model/study, theory or merely some text describing why this should be expected.
>
> In addressing the concerns about the foundational motivation for dormancy ratio minimization, it is important to clarify the relationship between neuron dormancy and exploration. Our approach is not based on the premise that a lower neuron dormancy directly leads to more exploration. Rather, we focus on using increased exploration as a means to reduce neuron dormancy. The correlation we observe is between neuron dormancy and activity level.
> When the policy network exhibits a high dormant ratio, this is typically associated with less active behaviors of the agent. In such scenarios, the strategy is to enhance exploration to decrease the dormant ratio. This process aims to activate a broader range of neurons within the network, potentially leading to the discovery of more effective behaviors. Therefore, the paper emphasizes the use of exploration as a tool to modulate neuron dormancy, rather than suggesting a direct causal relationship between low dormancy and increased exploration.
>
> > 2. There are a few aspects of the proposed algorithm that are mildly worrying.
> First is the exploitation mechanism. I do not understand how exploitation is determined by a hyperparameter controlling a value update towards a state-value or an action-value.
>
> In Section 3, our focus on Expectile Regression and Exploitation is aimed at enhancing the initial skill acquisition phase in reinforcement learning, where the policy $\pi$ is often sub-optimal and $Q(s,a)$ underestimates the true state and action values. To counteract this, we introduce a value estimator, denoted as $V$, which is designed to converge more rapidly than the traditional $Q$-value. This faster convergence of $V$ allows for a more accurate and immediate estimation of the ground truth value, effectively addressing the underestimation issue inherent in early learning stages.
> This method is crucial for guiding the agent towards more efficient exploitation of learned behaviors and successes. By leveraging the improved estimations provided by $V$, the agent can more effectively exploit its past experiences to accelerate the learning of an optimal policy.
>
> > 3. The second thing is that the algorithm proposes three mechansism, each with a hyperparmeter that governs a schedule. This provides several degrees of freedom to improve upon the base algorithm, and no hyperparameter study is conducted. This is not good empirical science, but the empirical demonstration of success on hard problems does at least hold promise for further work.
>
> Thank you for your feedback regarding the hyperparameters in our algorithm. We acknowledge the need for a clearer explanation of the motivation behind our scheduling approach. Our method employs a relaxed version of the “0-1” combination, where the hyperparameters dictating the extent of this relaxation are not overly sensitive. We have conducted all our experiments using this consistent set of newly added hyperparameters. The results in the paper have been updated to reflect these experiments, demonstrating the robustness of our approach across different settings.
>
>
> > 4. Section 3 (Language surrounding Dormant Ratio, specifically beginning of 3.2): While you demonstrate the empirial obervation that low neuron dormancy is correlated with meanginful high-level behavior, it is overclaiming to say that it is essential. I can imagine an approach to construct a neural network with an arbitrarily high dormancy ratio by encapsulating any policy within a much larger but inactive network. Thus, it is not necessarily the case that high dormancy ratio translates to less meaningful behavior, but that it can be correlated with it during training.
>
> Thank you for your insight. We agree that if we don't assume the network capacity, indeed, we could arbitrarily embed an active network (without a high dormant ratio) into an arbitrarily large network with an arbitrarily large dormant ratio. Here, we have the basic assumption that the network capacity (i.e., the size of the neural network) is fixed so that we cannot simply embed it into an arbitrarily larger network. We incorporate your insight and add this into the first sentence of 3.2.

---

> > ### Comment · Reviewer_jJ9Z · 2023-11-21
> >
> > Thank you for your detailed rebuttal. I particularly note that you made an effort in further improving your experiments by comparing against more baselines, and investigating the hyperparameter dependence. Regarding your reply:
> >
> > > Dormancy Ratio, Acitivity and Exploration
> >
> > Thank you for clarifying, it is a subtle point that exploration is being used to minimize the dormant ratio, rather than the other way around. The connection between network activity and meaningful behaviors is still unclear, but I do understand that it is an empirical observation that they are correlated. Furthermore, it serves as good motivation for this paper while leaving interesting directions open for future work.
> >
> > > Expectile Regression and Exploitation
> >
> > I agree that a state value function can be easier to learn, and that using and learning a state value function may allow you to learn a better policy quicker. While this is interesting, the use of the term exploitation here can still be confusing in my opinion. Is this a common terminology? I would encourage the authors to consider another term, or discuss how your usage of the term relates to the typical action-selection definition. (Exploitation is usually in regards to greedy action selection given a value function, whereas exploitation in your case is extracting more information from available experiences. It is an interesting distinction, and worth clarifying.)

---

> > > ### Author Response · Authors · 2023-11-21
> > >
> > > > I agree that a state value function can be easier to learn, and that using and learning a state value function may allow you to learn a better policy quicker. While this is interesting, the use of the term exploitation here can still be confusing in my opinion. Is this a common terminology? I would encourage the authors to consider another term, or discuss how your usage of the term relates to the typical action-selection definition. (Exploitation is usually in regards to greedy action selection given a value function, whereas exploitation in your case is extracting more information from available experiences. It is an interesting distinction, and worth clarifying.)
> > >
> > >
> > > In response to your comment on the use of the term "exploitation," we acknowledge that the term might not be immediately clear in the context of our paper. Our usage of "exploitation" adheres to the traditional understanding in reinforcement learning[1][2], aligning with Richard S. Sutton's definition in "Reinforcement Learning: An Introduction". Sutton describes exploitation as "exploit what it already knows in order to obtain reward". In line with this, we use the term to refer to the agent's actions that leverage gathered information to maximize rewards, a concept directly related to the role of the state value function in our work.
> > > We appreciate your suggestion to reconsider our terminology. To clarify, our paper's use of "exploitation" specifically pertains to extracting more value from available experiences, in contrast to the typical action-selection definition. We agree that this distinction is significant and warrants further clarification. Therefore, we will update our paper to better elucidate our use of the term and its relevance to our method.
> > >
> > > References:
> > >
> > > [1] Li M, Huang T, Zhu W. Clustering experience replay for the effective exploitation in reinforcement learning. Pattern Recognition, 2022, 131: 108875.
> > >
> > > [2] Schäfer L, Christianos F, Hanna J, et al. Decoupling exploration and exploitation in reinforcement learning. ICML 2021 Workshop on Unsupervised Reinforcement Learning. 2021.

---

> ### Author Response · Authors · 2023-11-19
> **Response to reviewer jJ9Z (2)**
>
> > 5. Section 3 (Expectile Regression and Exploitation): This section seems to build off recent work, and I am not familiar with "blended exploitation and exploration". However, I do not see how the proposed method has anything to do with exploitation. Exploitation and exploration are fundamentally about control and action, but this section is primarily about the target of a temporal difference method. But even putting that aside, I do not see why placing higher weight on the state value function should empahsize exploitation, or why placing higher weight on the action value function should emphasize exploration.
>
> Thank you for raising this important point. We acknowledge that our use of the term "exploitation" might require further clarification. In our context, "exploitation" refers to a strategy that could allow the RL agent to quickly exploit its serendipitous, unexpectedly successful experiences from its exploration process. The key insight that we observe is that, especially in the early stages of training, the replay buffer could contain scarce high-quality episodes (i.e., episodes with high reward) that the agent has encountered through exploration. In this training stage, $\pi$ is suboptimal, and the $Q$-value is often underestimated due to insufficient exploitation of high-quality samples in the replay buffer. Thus, to mitigate this underestimation, we introduce a value estimator $V$, which estimates the high expectile of $Q$, designed to converge more rapidly than $Q$-values. Thus, this mechanism allows the RL agent to quickly exploit its historically successful trajectories **without introducing additional overestimation**. (This is systematically shown in Ji et al. [1])
>
>
>
> > 6. Section 3 (Hyperparameters): The method involves three components, each with at least one hyperparameter. This gives your proposed algorithm a number of additional degrees of freedom over the base algorithm and the other baselines. Looking at the experiments, there is an ablation study for each individual component but no results on the sensitivity to the various hyperparameters. It would have been good to explore this, at least in one of the simpler environments.
>
> Thank you for pointing this out. This indeed deserves further clarification. In Appendix D.2, we now present the performance of DrM across various tasks using a single, unified set of hyperparameters, as opposed to domain-specific ones. We observed that DrM off-the-shelf generally performs well across all tasks with this single set of hyperparameters. However, for certain tasks, finetuning specific DrM hyperparameters does yield additional performance improvements. It's also important to note that the hyperparameter tuning required for DrM is minimal. The adjustments we made were limited to the maximum perturbation rate, selecting from $0.6$ and $0.9$, and modifying the exploitation expectile value by selecting from $0.7$ and $0.9$.
>
>
>
> > 7. Section 4 (Dormant ratio analysis): Validating the fact that the dormancy ratio is indeed minimized vs the base algorithm is interesting. One thing that would strengthen these results further is to show that the other baselines (A-LIX, TACO) are similar to DrQ-v2 in that they do not minimize the dormancy ratio on at least some problem. It would also be interesting to investigate why there is some periodicity in the dormancy ratio such as in humanoid run and manipulator.
>
> Thank you for your insightful comments regarding the dormant ratio analysis in Section 4. The periodicity observed in tasks such as humanoid run and manipulator should be due to the periodic perturbation of weights in our algorithm. In response to your suggestion, we have carried out further experiments with the A-LIX and TACO baselines to examine their impact on the dormant ratio. The results of these experiments are now included in the updated Figure 9. (Footnote: We re-executed all four algorithms, including DrQ-v2 and DrM, to regenerate Figure 9, as the original dormant ratio logging files were lost.) These additional experiments show that, similar to DrQ-v2, these baselines do not effectively minimize the dormant ratio to the extent observed with DrM.
>
> Reference:
> - [1] Ji et al. Seizing Serendipity: Exploiting the Value of Past Success in Off-Policy Actor Critic

---

> ### Author Response · Authors · 2023-11-19
> **Response to reviewer jJ9Z (3)**
>
> > 8. Section 4 (Ablation study): I appreciate the effort to ablate your algorithm in Adroit, a relatively complex environment. One thing that would help is to show whether each component of DrM is faithful to its motivation. For example, the exploration mechanism could be shown to have a large influence on performance in environments that require exploration and less influence on environments that do not require as much exploration. This would also serve as an opportunity to further elaborate on what the exploitation mechanism accomplishes.
>
> Thank you for your question regarding the ablation study. We acknowledge the value of your suggestion to demonstrate how each component of DrM aligns with its intended purpose. However, in practice, our empirical observations indicate that all three mechanisms are integral to DrM’s success. We find that exploration and exploitation are intrinsically interconnected; it is rare to encounter tasks that exclusively demand exploration or exploitation. In reality, a balanced and efficient blend of both exploration and exploitation is necessary for optimal performance.
>
> > 9. Section 2 (visual RL): Is it necessarily the case that your setting is POMDP and that the observations provided to the agent lack some information? While this can be true in some settings, the environment is usually constructed so that the observatiosn do include all necessary information (e.g. concatenation of frames in Atari).
>
> Indeed, our experiments have been conducted with frame concatenation to deal with partial observability. Here, adopt the convention of visual RL for both continuous control and Atari games, where the observation is a stack of 3 RGB images of the consecutive state following DrQ-v2. The observation after frame stacking should include all necessary information to make a decision without using a recurrent/transformer-based policy network to incorporate the history.
>
>
>
> > 10. Section 2 (Dormant Ratio): You use the term "Linear layer" in both definitions, but I am not sure whether this is required in the definition or what it is that you mean. Does it mean the last layer that usually maps the penultimate activations to the output?
>
> We clarify that the term "Linear layer" used in Section 2 refers to all linear/fully connected layers ('nn.Linear'), not just the last output-mapping layer. To avoid confusion, we've updated our paper to replace "Linear layers" with "Fully Connected layers." This change accurately represents our model, which includes both convolutional neural networks and fully connected networks, with the dormant ratio calculations considering all fully connected layers.

---

### Author Response · Authors · 2023-11-19
**Response to all reviewers**

We sincerely thank all the reviewers for your thoughtful feedback and constructive suggestions. We are encouraged by the reviewers' recognition of the strengths in our work, including the innovative insight into dormant ratio and the application of dormant ratio as an effective intrinsic indicator to manage the exploration-exploitation tradeoff ("provides a novel lens for the phenomenon" jJ9Z, "a promising direction to tackle the exploration-exploitation dilemma" 2E41, "The idea of strategically exploring based on properties of an agent's representation is a new and exciting approach which I think could be applied more broadly in RL and could be really useful to the community." R5aS), broad empirical evaluation across multiple domains ("evaluates on more than one domain" R5aS, "The authors have considered three separate domains, with multiple tasks from each." Jxw3) and strong empirical results compared with existing visual RL baselines ("the improvements seem statistically significant" jJ9Z, "significantly outperforms existing methods" 2E41, "significantly outperforms the baselines" R5aS, "DrM is the clear winner on all of the tasks" Jxw3).

In response to the valuable feedback from reviewers, we have conducted extensive additional experiments and provided additional discussions to integrate your insightful comments into our work to further enhance the overall quality of our study. Here, we briefly outline the major updates to the revised submission for reviewers' reference.

1. **Results of DrM with a single set of hyperparameters**: In Appendix D.2, we present the results of DrM on 9 tasks that used a different hyperparameter setting, compared with the default setting using a single unified set of hyperparameters. In general, we find that applying DrM with the default hyperparameter setting off-the-shelf achieves decently great performance, while additional hyperparameter tuning could further improve its performance. Additionally, in the text of the updated manuscript, we also emphasized that the additional hyperparameter tuning efforts should be minimal since we only need to adjust two parameters, each selecting from two candidate values.
2. **Additional Ablation Study**：In Figure 10, we have added an additional baseline into our ablation study, specifically fixing the hyperparameters of the three components without considering the dormant ratio. This analysis reinforces the significance of the dormant ratio, illustrating that the combination of the three components alone does not capture the full utility of our approach.
3. **Comparison against intrinsic exploration techniques**: In Appendix E, we compare DrM with RND, an exploration approach with intrinsic rewards. DrM outperforms RND across all three tasks, which further underscores the significance of the proposed approach.
4. **Comparison with ReDo**: In Appendix F, we present an additional comparison between DrM and ReDo, which solely resets the weights of dormant neurons, different from the shrink and perturb mechanism in DrM. Figure 14 clearly demonstrates the significant performance gains of DrM over ReDo, underscoring the effectiveness of our design choice.
5. **Enhanced Baseline Comparisons for Dormant Ratio Visualization**: To more comprehensively illustrate DrM's effectiveness in minimizing the dormant ratio of its policy network, we have revised Figure 9. This updated figure now includes additional visualizations of the dormant ratio for baseline models TACO and A-LIX, which were not part of the original manuscript. We observe that as training progresses, the dormant ratio of DrM rapidly decreases. In comparison, other exisiting baselines all fail to effectively reduce the dormant ratio.
6. **Further behavioral analysis**: Beyond the Hopper Hop task illustrated in Figure 1, we have included additional qualitative analysis and visualization of the agent's dormant ratio and corresponding behaviors on our anonymous project webpage https://drm-rl.github.io/asset/dormant.mp4.

Thanks again for the insightful comments and feedback provided by all reviewers. Should there be any further questions, we are more than willing to address them.

---

### Meta-Review · Area_Chair_paFk · 2023-12-05

**Metareview:**

This paper presents an algorithm which leverages the fraction of dormant units in the network to guide exploratory behaviour. The resulting method demonstrates significant improvements over baseline algorithms that use naive exploration strategies on a variety of challenging robotic manipulation domains. All authors agree that the paper is ready for publication, thus I’m recommending its acceptance. I acknowledge that the authors have done a great job addressing the reviewers' concerns due the discussion phase. I encourage the authors to go over the reviews one more time and address, in the paper, any points that were not addressed yet, as all the reviews provided meaningful feedback to the authors.

**Justification For Why Not Higher Score:**

All reviewers think the paper should be accepted but no one wanted to fight really hard for this paper. Many requests were made and the authors addressed most of them, but for a higher score I'd expect a more fundamental understanding of the proposed techniques. More than one reviewer mentioned they would like to understand better why the proposed approach works so well.

**Justification For Why Not Lower Score:**

Having established that this paper should be accepted for publication, the question is whether it should be a poster or spotlight talk. My recommendation for spotlight instead of poster is based on the fact that the paper tackles an important problem and it does so by incorporating techniques from another growing field: continual reinforcement learning. I find it particularly useful to give more visibility to some of those papers.

---

### Decision · Program_Chairs · 2024-01-16

Accept (spotlight)